



# Development of the SiMPLE-PAS: A low-cost, 3-wavelength photoacoustic spectrometer for aerosol absorption

Ashley M. Scott[1], Charles A. Wise[1,2], Ryan P. Poland[2], Anna D. Jordan[1], and D. Al Fischer[1]

[1]Department of Chemistry and Physics, Western Carolina University, Cullowhee, NC, USA
[2]Department of Chemistry, University of Georgia, Athens, GA, USA

**Correspondence:** D. Al Fischer (dfischer@wcu.edu)

**Abstract.** Photoacoustic spectroscopy (PAS) has become a common method for measuring aerosol absorption, and is one of the few techniques capable of directly measuring absorption by suspended aerosol particles at ambient concentrations. When multiple wavelengths are used, PAS provides a way of measuring the absorption Ångström exponent, and, when combined with a scattering or extinction method, provides a measure of the aerosol single scattering albedo, and both AAE and SSA are important parameters in climate models. Despite this utility, few commercial PAS instruments are available and no multi-wavelength commercial instruments are currently available. Thus, most extant PAS instruments are custom-built and therefore come with considerable cost and development time and require access to machine shops capable of fabricating the needed components. The goal of this work was to provide a blueprint for a low-cost, multi-wavelength PAS for measurement of the aerosol AAE both in the laboratory and in the field. In an effort to create an instrument with a low barrier to entry, we aim to use low-cost, readily available components and open-source options wherever possible. In this manuscript, we present the SiMPLE-PAS, a single-pass, multi-wavelength, portable, and low-expense photoacoustic spectrometer that uses low-cost electronics and a 3D-printed cell to meet these design goals. The instrument has a total bill-of-materials cost on the order of $500 USD. The instrument is, to the best of our knowledge, the first 3D-printed PAS for aerosols and the lowest-cost PAS to date. The instrument performed well in laboratory validation experiments, and showed good agreement with measurements of aerosol absorption by the previously developed MultiPAS-IV instrument when co-located at the second Georgia Wildland Fire Simulation Experiment (G-WISE 2) during April 2025. The instrument shows competitive detection limits of 0.63, 1.99, and 0.55 $\mathrm{Mm}^{-1}$ for the blue, green, and red channels (10-minute, 2-$\sigma$), respectively, that will allow it to measure both ambient and laboratory-generated aerosols. The SiMPLE-PAS therefore provides a low-cost, accessible photoacoustic spectrometer that offers to lower the barrier to entry for groups wishing to measure aerosol absorption, whether in the laboratory or in the field.

## 1 Introduction

This manuscript presents the initial development of the SiMPLE-PAS, a single-pass, multi-wavelength, portable, and low-expense photoacoustic spectrometer designed to measure aerosol absorption and the aerosol absorption Ångström exponent (AAE). The instrument relies on the technique of photoacoustic spectroscopy (PAS) to directly measure the absorption coefficient of suspended aerosols at three wavelengths using inexpensive blue (450 nm), green (515 nm), and red (665 nm) laser



diodes. PAS is one of the few absorption techniques with the sensitivity required to measure absorption of light by suspended aerosols that is not also sensitive to scattering. (Krzempek, 2019) This allows measurement of the aerosol single scattering albedo (SSA) when combined with an extinction or scattering method. When multiple wavelengths are used, PAS allows retrieval of the AAE, which describes the wavelength dependence of the aerosol absorption spectrum. Such measurements of SSA and AAE are vital for understanding Earth's climate. (Moosmüller et al., 2009; Li et al., 2022)

In the atmosphere, absorbing aerosols are typically composed of black carbon (BC), brown carbon (BrC), or a mixture thereof, with an absorption spectrum following the form of $b_{abs} = \beta \lambda^{-AAE}$, where $b_{abs}$ is the absorption coefficient of the aerosol, $\lambda$ is the wavelength, and $\beta$ is the pre-exponential scaling factor. (Moosmüller et al., 2009) These categories are largely defined by their spectroscopic characteristics (not chemical composition), with black carbon having an AAE of 1.0 and brown carbon having an AAE of $>> 1.0$. (Moosmüller et al., 2011; Saleh et al., 2018) Instruments with at least 2 wavelengths

can retrieve the aerosol AAE using a linear fit to log-transformed data, although it is typically better to have at least 3 or 4 wavelengths for separating the individual contributions of BC and BrC to the overall absorption spectrum. (Zhang et al., 2016; Nakayama et al., 2015). Fischer and Smith (2018a) developed a 4-wavelength, multi-pass PAS called the MultiPAS-IV and Saleh et al use a 3-wavelength derivative of this instrument called the MultiPAS-III (e.g. Yu et al., 2021; Cheng et al., 2021). More recently, Schnaiter et al. (2023) developed the PAAS-4$\lambda$, which used a single-pass PAS cell but yielded similar detection

limits to multi-pass instruments. And even more recently, a 3-wavelength cantilever-enhanced PAS (i.e. QE-PAS) has been developed for aerosol absorption (Karhu et al., 2025).

   Despite its use as one of the only direct, online methods of measuring aerosol absorption, few commercial PAS instruments exist and many PAS groups use custom-built instruments. (Upadhyay et al., 2025; Nakayama et al., 2015) There is therefore a significant up-front expense in terms of both financial cost and time cost to develop an instrument. Although numerous PAS

designs have come to exist with varying levels of complexity, in principle a PAS is composed of relatively few components and could be built relatively cheaply. The required components are (1) a light source, typically a laser, capable of being modulated at a frequency of several kHz; (2) a sample cell that consists of an acoustic resonator (usually) with windows on either end; (3) a microphone and preamplifier to detect and amplify the signal; and (4) a data acquisition card capable of digitizing the relatively slow (several kHz) audio signal produced by the microphone. Often, instrument designers will also include a photodiode or

power meter at the back of the sample cell to monitor changes in source power in real-time (e.g. Ajtai et al., 2010; Nakayama et al., 2015; Linke et al., 2016; Fischer and Smith, 2018a; Yu et al., 2019; Karhu et al., 2022; Schnaiter et al., 2023). Despite these things, low-cost has rarely, if ever, been realized in practice.

   Many options now exist for low-cost electronic components (e.g. development boards such as Arduino and Raspberry Pi Pico) and manufacturing techniques (e.g. fused deposition modeling, FDM '3D printing'), but few have been applied to PAS.

Recently, Haedrich et al. (2025) presented a "low-cost optoacoustic black carbon sensor" based on quartz-enhanced PAS (QE-PAS) and designed to measure ship emissions via on-board deployment. This was based on the QE-PAS developed by Stylogiannis et al. (2021). QE-PAS differs from traditional PAS in that the signal is detected with a quartz tuning fork instead of a microphone. Therefore, while this instrument employs low-cost consumer electronics (i.e. Arduino), it still requires a custom-machined sample cell and further requires a custom quartz tuning fork or a commercial QE-PAS cell. Although no





price is given in the manuscript, the instrument is described as "relatively lower cost" by the authors and it seems reasonable
to assume the instrument costs in the range of at least several thousand dollars based on the cost of commercial QE-PAS cells
(which are roughly $4000 USD via Thor Labs at the time of this writing and do not include light sources or electronics).
Keeratirawee et al. (2022) developed low-cost circuitry for PAS, but do not present a corresponding low-cost cell design and
present applications to gases only, not aerosols, and only at a single wavelength. Kapp et al. (2019) developed an LED-based

photoacoustic sensor for nitrogen dioxide. The authors do not describe this instrument as low-cost, but did use a low-cost
microphone and home-built preamplifier along with an innovative "T-cell" resonator design that allowed a low-cost LED to be
used as the light source. This instrument was also manufactured using additive manufacturing in the form of direct metal laser
sintering (DMLS), which allowed for a unique and easy to manufacture cell, although DMLS machines themselves are quite
expensive and not as widely available as other additive manufacturing equipment (e.g. FDM 3D printers). Helmholtz-style

PAS cells have also recently been explored for gas-phase measurements. Such cells use alternative resonator designs that have
features that make them potentially amenable to low-cost designs, including short, large-diameter sample cells that would allow
for use of LED light sources. For example, Fu et al. (2024) used FDM 3D printing to build a Helmholtz-based PAS for trace gas
detection, and Zhang et al. (2024) developed a differential Helmholtz PAS with a short, large diameter that could accommodate
a laser multi-pass cell, although it would also be amendable to LEDs. However, most of these studies have largely focused on

the development and description of the Helmholtz resonators and there have been no applications to aerosols and no apparent
attempt to use low-cost components for the light source or detection/DAQ portions of the instrument. In short, no PAS has
combined all the features needed for a truly low-cost PAS, and those that come close have not been applied to aerosols and/or
measure only at a single wavelength.

        The objective of this work is to explore the viability of using low-cost electronics and 3D printing to construct a multi-

wavelength PAS for aerosol absorption, with the ultimate goal of presenting a simple, accessible, low-cost design for an
aerosol PAS using open-source hardware and software wherever possible. Although several groups have made progress toward
low(er)-cost PAS instruments or components thereof (Haedrich et al., 2025; Keeratirawee et al., 2022; Kapp et al., 2019), no
group has presented a low-cost multi-wavelength PAS. Further, existing manuscripts largely omit details about how to apply
low-cost methods to scientific instrumentation, such as the steps used to create gas-tight sample cells using FDM 3D printing.

Here, we present a portable, sensitive, single-pass, laser-based, 3-wavelength PAS that that measures aerosol absorption at
450, 515, and 665 nm that can be constructed for a materials cost of roughly $500. We further discuss the considerations and
cautions that go into developing a gas-tight instrument with FDM 3D printing. We anticipated that building a PAS for such
low cost would inherently limit sensitivity; although this is true to an extent, the SiMPLE-PAS has detection limits that are
competitive when compared to much more expensive instruments and exhibits a large dynamic range. Together, these features

make it suitable for use in measurements of ambient atmospheric aerosols where concentrations are often very low and in lab
campaigns where concentrations are often very high. In this manuscript, we present the design of the sample cell, electronics
and data acquisition hardware, data acquisition software, and portable case and mounting system, with detailed discussion
of considerations taken and tradeoffs made to make the instrument low-cost. We then describe calibration of the instrument
with $NO_2$ and validation with gas-phase measurements of the ozone absorption cross section, discuss limits of detection, and

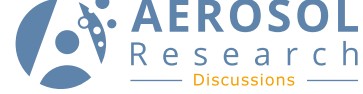



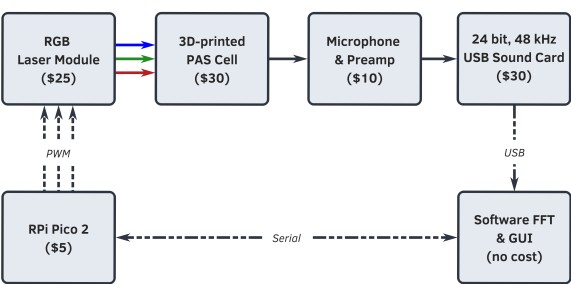

**Figure 1.** Functional block diagram of the SiMPLE-PAS, including rough cost estimates for each item. A Raspberry Pi Pico 2 development board is used to control an RGB laser module that illuminates the 3D-printed PAS cell at 3 distinct frequencies (one for each wavelength); the signal is digitized with a USB sound card, and an FFT is performed on the signal using custom software written in Julia, which also allows for user control of the lasers and other PAS components. A full cost breakdown is provided in the supplemental information.

finally demonstrate the capabilities of the instrument to measure aerosols with deployment at the G-WISE 2 (Georgia Wildland Fire Simulation Experiment 2) laboratory campaign, where the instrument was co-located with the MultiPAS-IV (Fischer and Smith, 2018a) instrument for intercomparison.

## 2   Instrument description

The SiMPLE-PAS is a low-cost, single-pass, 3-wavelength, laser-based photoacoustic spectrometer that uses a sample cell
manufactured with FDM 3D printing. To the best of our knowledge, the SiMPLE-PAS is the first multi-wavelength PAS and first aerosol PAS manufactured with 3D printing and the first PAS under $1000 (USD), making it orders of magnitude less expensive than most PAS instruments. Figure 1 shows a functional block diagram of the SiMPLE-PAS, including rough cost estimates for each item (a full cost breakdown provided in Fig. S6). It is composed of a low-cost red, green, blue (RGB) laser module controlled by a Raspberry Pi Pico 2, a 3D-printed cell, a custom microphone and preamplifier board, a 24-bit USB
sound card, and open-source software written in the language Julia. We will describe these components in this section, with a special focus on steps unique to developing a low-cost and 3D-printed PAS.

### 2.1   Measurement principle

The SiMPLE-PAS relies on the technique of photoacoustic spectroscopy (PAS), which has been described in detail in various reviews and tutorials. (Bell, 1880; Miklós et al., 2001; Dumitras et al., 2007; Palzer, 2020; Upadhyay et al., 2025) In the
SiMPLE PAS, as with most PAS instruments, a laser is directed into a sample cell, which is composed of an acoustic resonator of length $\lambda/2$ (where $\lambda$ represents the wavelength of a waveform at the resonant frequency) with a $\lambda/4$ resonator on either end to cancel background noise. Here, the main $\lambda/2$ resonator has a nominal length of 100 mm. The sample is carried into the resonator by a bath gas, typically air, nitrogen, or oxygen depending on the analyte, and light absorbed by the sample



is converted to heat via non-radiative decay and ultimately transferred to the bath gas. (Cotterell et al., 2019, 2021) As a first approximation, one can surmise that this increase in temperature leads to an increase in pressure via the ideal gas law ($PV = nRT$). Thus, when the light source is modulated on and off at the inherent resonant frequency of the sample cell, a standing wave forms wherein the amplitude of the waveform is proportional to the amount of light absorbed by the sample. The cell is designed with an acoustic resonant frequency (in our case roughly 1600 Hz), such that a microphone placed at the center of the sample cell (i.e. the antinode of the standing wave) can then detect the pressure (sound) wave and the volume of the detected sound is proportional to the amount of light absorbed. For a single wavelength, the sound wave resultant on the microphone can be described by:

$$S_\lambda = P_{0,\lambda} M \left( C_{\text{cell}} \cdot \sigma_{\text{abs},\lambda} \cdot N + A_{\text{b},\lambda} \right) \tag{1}$$

Here, $S$ is the signal detected by the microphone (units: V) at wavelength $\lambda$, $P_0$ is the power of the incident light (W), $M$ is the sensitivity of the microphone (V Pa$^{-1}$), $C_{\text{cell}}$ is the PAS cell constant (Pa M W$^{-1}$), $\sigma_{\text{abs}}$ is the absorption cross section of the analyte (m$^2$), $N$ is the number density of the analyte (m$^{-3}$), and $A_{\text{b}}$ is the background signal (Pa W$^{-1}$). (Ajtai et al., 2010; Bozóki et al., 2011; Schnaiter et al., 2023) The cell constant is dependent on the geometry of the sample cell and the bath gas, according to:

$$C_{\text{cell}} = \frac{(\gamma - 1)QG}{2\pi f_m A_{\text{res}}} \tag{2}$$

with $\gamma$ equal to the ratio of specific heat constants of the bath gas, $f_m$ equal to the resonant frequency of the cell (Hz), $A_{\text{res}}$ representing the cross-sectional area of the cell (m$^2$), $Q$ being the unitless quality factor of the resonator, and $G$ being the unitless 'geometric factor' that can range from 0 to 1 and accounts for the spatial overlap (or lack thereof) between the laser beam and acoustic mode of the cell. Ideally, all terms in Eq. (2), and thus $C_{\text{cell}}$, should be constant with wavelength. But for the multi-wavelength cell described here, the lasers are each modulated at a different frequency and run simultaneously, such that the waveform detected by the microphone represents the sum of Eq. 1 at three separate wavelengths, where $f_m$ is set uniquely for each wavelength. Although the modulation frequencies are close enough that $Q$ varies little from one wavelength to another and we work to keep $G$ the same for all wavelengths, we measure $C_{\text{cell}}$ independently for each wavelength during calibration (as discussed in the calibration section below).

## 2.2 3D-printed photoacoustic cell

As built, the photoacoustic cell used in this work consists of a 101 mm long by 4.6 mm inner diameter stainless steel resonator held in a 3D-printed plastic body with 50 mm by 35 mm inner diameter buffer volumes on either end. The body of the cell and buffer volumes are manufactured from PET-G (polyethylene terephthalate) using a Prusa i3 MK3S+ (Prusa Research). Briefly, FDM 3D printing works by extruding plastic filament through a hot nozzle, and the placement of the nozzle is controlled by G code, such that the 3D printer can build up a part layer by layer by placing the nozzle where plastic is desired; each layer



of plastic is typically 0.2 mm thick. FDM 3D printing was chosen as the manufacturing method for its low cost and relatively
widespread availability, with many academic institutions having maker spaces that contain 3D printers, many affordable 3D
printer options available to consumers, and online vendors who will 3D print components for a relatively low cost. But 3D-
printed parts present multiple challenges when incorporated into a photoacoustic spectrometer. First, and most importantly,
parts made with FDM 3D printing are typically not gas-tight by default. Because typical calibration methods for PAS rely on
pressurized, toxic gases ($NO_2$, $O_3$), this is a major concern and thorough testing must be taken to ensure the instrument will
not leak gases into the lab. Similarly, any leaks in the instrument could pose experimental issues when under vacuum, such is
the case when sampling ambient aerosols or when sampling from a common sampling line in a laboratory campaign. Second,
FDM 3D printed parts are typically composed of 'solid' inner and outer shells that are typically only 3-5 layers thick, with
most of the inner volume of the part composed of 'infill' with a default density of 15-20% plastic in a honeycomb or rectilinear
pattern with empty space in between; for PAS, this creates a low-density part through which sound can easily travel and cause
interference in the signal. Finally, tolerances are often poor in FDM 3D printing and parts contain layer artifacts, such that the
geometry of a particular feature may depend on the orientation in which the part was printed. For example, a cylinder (or PAS
resonator) printed on end will have a circular cross section (because the nozzle can easily move in a circle) while a cylinder
printed on its side will have an *approximately* circular geometry because it must be built up of 0.2 mm layers. In PAS, this
is equivalent to a rough resonator and leads to a low $Q$ and therefore low sensitivity. In another example, surfaces printed in
contact with the build plate or as the top surface will have a relatively smooth surface (assuming the build plate is smooth),
while surfaces on the sides of parts will be rough; this becomes an important consideration when printing o-ring grooves and
other sealing surfaces.

Here, we have developed solutions to each of these problems. Figure 2 and Fig. S2-S3 show 3D CAD models and photos of
the hardware that composes the PAS cell, which is printed in 3 pieces: (1) the central resonator and (2 and 3) the two buffer
volumes, which are are identical to each other. As shown in the cross section in Fig. 2C and photos in Fig. S3, each part is
printed as a plastic shell that defines the overall geometry of the cell, but the majority of each part is empty space. To help
facilitate a sealed part, we add additional perimeter layers, use a higher infill of 50%, and set the 3D printer to over extrude each
layer slightly. Then, during the first step of assembly, we coat all parts with a spray-on acrylic coating (Minwax Polycrylic)
to fill any voids left between layers of the 3D print. We note that there are commercial products designed to coat 3D prints
and make them gas-tight (e.g. Dichtol by Diamant Polymer Solutions), but we were unable to procure these products in the
United States during this study. We instead chose a widely available, low-cost consumer product. For the buffer volumes, we
fill the empty space designed into the parts with a slurry of calcium sulfate in water, which we allow to harden for several
days. We do the same for the main resonator part, but we first press a stainless steel tube of the desired length and diameter for
the resonator into the 3D printed part; we use 305 stainless steel 6 gauge thin-walled hypodermic tubing (5.2 mm O.D., 4.6
mm I.D.) for this and wet sand the inside to a near mirror finish prior installation. The purpose of this tube is to (1) provide
a smooth surface with clearly defined diameter for the resonator, which improves $Q$, (2) provide a metal surface to minimize
particle wall loss in the resonator, and (3) provide a fully sealed surface inside the resonator to minimize leaks. After being
pressed in, the joints where the tubing meets the plastic part are sealed with cyanoacrylate adhesive (Maxi-Cure, Bob Smith

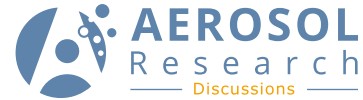



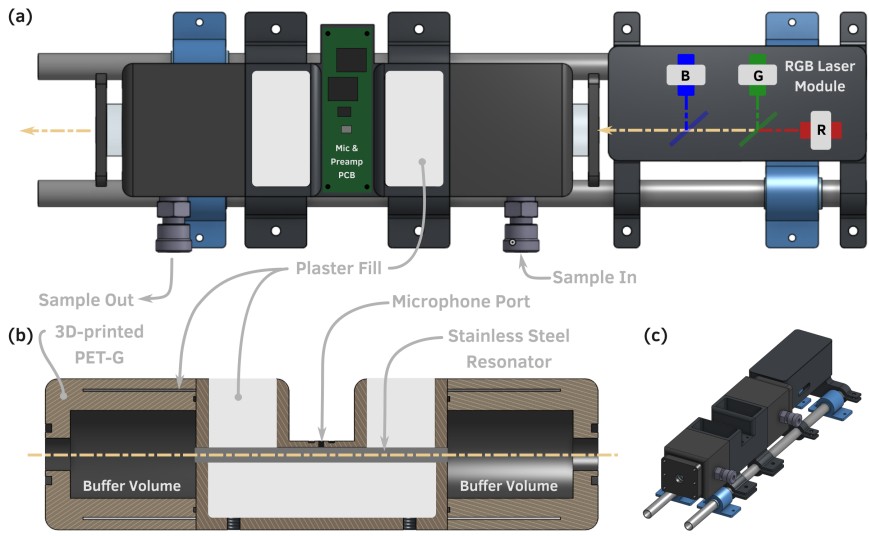

**Figure 2.** Hardware components of the SiMPLE-PAS. (a) Top view of the SiMPLE-PAS. (b) Cross section (side view) of the PAS cell showing the internal cell geometry including plastic shell, infill areas that are filled with calcium sulfate, and the stainless steel resonator tube. (c) Isometric view of the SiMPLE-PAS (infill regions shown empty). All components are mounted on a carbon-fiber rail system that screws into a rugged case via acoustic isolation mounts. Photos of the laser module, PAS, and filling process are given in the supplemental information.

Industries, Inc; we note epoxy could also be used if needed). Once cured, the paster infill provides a dense, solid part that is
more resistant to interference from external sounds. Calcium sulfate was chosen because it is low-cost, widely available, and once cured has a density similar to that of aluminum, the material from which many PAS cells in the literature are machined. After the curing, the plastic inlet and outlet holes on the buffer volumes are tapped with a 1/4" NPT tap and a corresponding fitting is installed (e.g. 1/4" NPT PrestoConnect or Swagelok) using PTFE tape to create a removable seal between the fitting and the plastic part; it is important to tap the holes *after* filling with plaster to avoid splitting the part along layers of the 3D
print. Finally, the buffer volumes are affixed to the main resonator with cyanoacrylate adhesive, using a gluing groove that is designed into the parts, and a 1 mm diameter hole is drilled in the center of the main resonator where the microphone will be installed. The cell is capped on either end with 25 mm diameter by 5 mm thick AR-coated flat N-BK7 windows mounted on an o-ring (Thor Labs, WG11050-AB), although we note that those looking to reduce cost further could use a cheaper option here. A 3D-printed plate holds the window in place, and is mounted on the end of the cell via M3 threaded metal inserts that are
heat set into the plastic cell body. Two 1/4"-20 (or M6) screws are set into the bottom of the cell to allow mounting to standard optical posts.

The PAS cell and laser module (described below) are mounted on two 12.2 mm diameter carbon fiber rods using 3D-printed mounts. Carbon fiber was chosen for its low thermal expansion and light weight; each rod costs roughly $25 USD, so those wishing to reduce costs could likely replace these rods with, e.g., half-inch PVC pipe. The carbon fiber rods are held on each

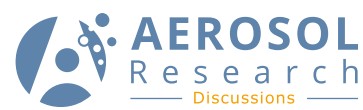

end by acoustic isolation mounts (HoldRite #1414509), which fasten the rods to the PAS case while isolating the PAS from
external sounds and vibrations. The isolation mounts are screwed directly into metal threaded inserts that are heat set into a
hard plastic case. The case serves to protect the PAS from dust and physical damage, protect users from exposure to laser
radiation, facilitate portability, and shield the PAS from external sounds. The inside of the case is lined with felt to provide
additional sound dampening. When fully assembled, the PAS has outer dimensions of 0.5 m x 0.4 m x 0.2 m and a weight of
roughly 5 kg.

   As noted above, it is imperative to leak test the PAS prior to use. We leak test each cell in two ways. First, we flow nitrogen
through the cell at roughly 300 sccm (the typical operating flow of the PAS) and measure the flow in and flow out. Any loss
greater than several sccm is considered a failing leak test. Second, we seal the outlet of the PAS and pressurize the cell to 10 psig
with nitrogen and then submerge it in water. Any bubbles that form are a sign of a failing leak test, and may be further useful
in pinpointing the location of the leak. Any cell that fails the leak test is not used and is either discarded or re-coated/re-glued
until it passes. We then ensure not to exceed 10 psig pressure in the cell during operation.

## 2.3   Laser module

The PAS resonator is illuminated with a low-cost RGB laser module (Fig. S1) that is designed as a replacement module for
laser projectors used in laser light shows. The module includes a control board that allows independent modulation of each
laser. These modules cost roughly $25 USD from online vendors (e.g. eBay, Amazon, Alibaba) at the time of this writing and
are sold generically as "500 mW RGB Laser Modules". They consist of three individual diode lasers and use dichroic mirrors
to make the beams colinear that are pre-aligned at the factory. Because these modules are commodity/consumer products, little
information is given about the laser specifications, but our measurements indicate wavelengths of 450 nm, 515 nm, and 665
nm (Fig. S1) with powers of 140, 32, and 100 mW, respectively. We enclose the module in a 3D-printed enclosure to protect
the optics from dust and other damage. The lasers are modulated at 3 distinct frequencies near the resonant frequency of the
cell using the PWM (pulse-width modulation) output of a Raspberry Pi Pico 2 with a 50% duty cycle. This allows the signals
for each wavelength to be demodulated using a fast-Fourier transform during data acquisition. Although the lasers specify a
24 VAC supply voltage, examination of the included control board reveals the 24 VAC is immediately passed through a bridge
rectifier and we surmise the lasers run on DC and that the 24 VAC supply is recommended because laser projectors may already
have a 24 VAC supply built in. Therefore, we choose to power them with a 12 VDC power supply and thereby eliminate the
need for a separate 24 VAC source. When using a 50% duty cycle at 12 VDC, the entire module draws roughly 150 mA,
consuming just under 2 W.

   The Raspberry Pi Pico 2 used to control the lasers costs roughly $5 USD and is based around the RP2350 chip, which
provides 12 PWM generators with 12-bit resolution on the duty cycle. We use 3 of the PWM channels to modulate the lasers
at a 50% duty cycle at three separate frequencies, such that the signal for each wavelength appears in an FFT at a distinct
frequency. Specifically, the green laser is modulated at the resonant frequency of the cell (i.e. $f_{m,515} = f_r$), the blue laser is
modulated at $f_{m,450} = f_r - 6$ Hz, and the red laser is modulated at $f_{m,665} = f_r + 6$ Hz. It was experimentally determined that
spacing closer than 6 Hz led to cross-talk between the signals for each color and increased noise. The laser controller runs





custom firmware written in Python that allows the user to set the modulation frequency and turn the lasers on and off via a

serial connection.

## 2.4   Preamplifier design

The detector in the PAS is a low-cost, high-sensitivity microphone, which produces a signal that requires amplification prior
to digitization. For cost savings and clean integration in the PAS design, we developed our own preamplifier board for the
SiMPLE-PAS. Although other groups have presented low-cost electronics for PAS, these electronics were based on lock-
in detection of a single wavelength, and are thus unsuitable for the multi-wavelength instrument here (Keeratirawee et al.,
2022). Figure 3 shows the preamplifier circuit designed for the SiMPLE-PAS. The microphone is a bottom-ported, surface-
mount (SMD) microphone (TDK ICS-40730), similar to that used by Kapp et al. (2019). The microphone is held on a custom
printed circuit board (PCB) that mounts directly to the PAS cell via 4 M2 threaded metal heatset inserts. A seal is formed
between the cell and PCB around the microphone port with an o-ring, with care taken to avoid any vias or other throughholes
within the sealing area of the PCB. The microphone has a nominal sensitivity of -32 dB (1 kHz, 94 dB sound pressure level)
and a signal-to-noise ratio of 74 dBA (20 Hz to 20 kHz, A-weighted) when operated in differential mode (as is done here),
yielding a noise floor of -106 dBV. The differential output from the microphone is sent to a passive, differential high-pass
filter with a cutoff frequency ($f_c$) of 1000 Hz to remove low-frequency noise from the signals. The filtered differential signals
are fed to a variable-gain instrumentation amplifier, which serves to cancel common mode noise while amplifying the signal
to a user-variable gain. The instrumentation amplifier converts the differential signal to a single-ended signal, which is sent
to an active high-pass Sallen-Key filter. That output is buffered prior to entering an active low-pass Sallen-Key filter. The
two active filters are designed with a gain of 2.25 to yield a relatively high $Q$ of 1.3, such that they amplify signal at the
resonant frequency of the cell while attenuating signals at other frequencies, as shown in the simulation results in Fig. S5.
Finally, the filtered signal is sent to an inverting amplifier, which we typically set to unity gain but could be arranged to
produce additional gain if desired. The output of the preamp board is AC-coupled to allow easy interface with off-the-shelf
audio components. The instrumentation amplifier is implemented via a dedicated instrumentation amplifier integrated circuit
(Texas Instruments INA118) in conjunction with a 5 kΩ potentiometer that functions as the gain resistor, $R_G$. The Sallen-
Key filters and buffer/inverting amplifiers are implemented using two dual low-noise, low-power precision op-amp ICs (Texas
Instruments MCP6292). All components are held on a custom PCB, which was manufactured by JLCPCB. The preamp requires
both a positive and negative supply; since the microphone has an upper supply voltage limit of 3.6 V, we use rail voltages of
+3.3 V and -3.3 V. For the data presented herein, these voltages are supplied with two standard benchtop linear DC power
supplies (Rigol DP711), although we have since switched to cheaper OEM power supplies with no apparent degradation of
performance; for this reason, and the fact that many labs may already have bench supplies on hand, the Rigol power supplies
are not included in the cost estimate for the SiMPLE-PAS.



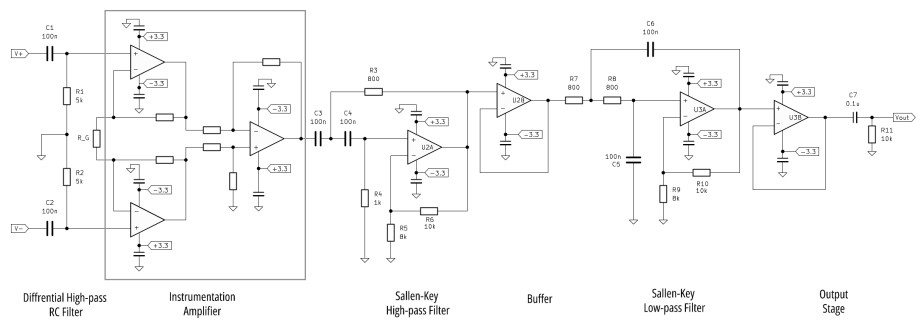

**Figure 3.** Equivalent schematic for the SiMPLE-PAS preamplifier. The differential output from the microphone is represented by $V+$ and $V-$ in the schematic. The instrumentation amplifier is implemented using a dedicated instrumentation amplifier IC, with only $R_G$ being supplied externally to the IC. U2 and U3 are dual op-amp ICs, such that the four op-amps for the filters and buffers require two ICs. Simulated output for the circuit is provided in the supplemental information.

## 2.5 Data acquisition and software

The output from the preamplifier is digitized at 48 kHz using a 24-bit USB sound card (World Star Tech), similar to Fischer and Smith (2018a). The sound card collects a 1-second waveform, which is then sent to a PC running a custom data acquisition program written in the language Julia (v 1.11) (Bezanson et al., 2017). The program intakes the 1-second waveform and performs an FFT on the data using the FFTW.jl library (Benzanson et al., 2025). The software then extracts the magnitude of the signal at each of the laser frequencies and saves each magnitude in a CSV file with a corresponding timestamp. It is necessary to integrate for 1 second to achieve 1-Hz resolution in the FFT, which is desired for deconvolution of the three closely spaced laser frequencies. The FFT itself and data processing and saving take an additional ~200 ms, leading to a total duty cycle of roughly 1.2 seconds; because this can vary from point to point based on what other tasks the processor is completing at any given moment, we throttle the program to a duty cycle of 2 seconds to maintain a consistent sampling rate of 0.5 Hz.

Although the program can be run from the command line, we use the Julia packages Pluto and PlutoUI to create a basic graphical user interface (GUI) for the SiMPLE-PAS. Pluto is a notebook environment developed specifically for Julia. It differs from other notebook environments (e.g. Jupyter) in that notebooks created with Pluto are reactive – that is, the notebook tracks execution order (such that the top-to-bottom order that code is written in does not matter) and the entire notebook updates automatically each time the user changes a variable; PlutoUI is an additional package that provides input/UI widgets such as buttons, sliders, file pickers, etc. that can be used to trigger code execution. (van der Plas et al., 2025) Although, as best we can tell, Pluto was developed as a tool to teach programming and share data analysis files in an interactive format, the features mentioned above allow for the creation of basic instrument GUIs. With the available widgets, order-agnostic code, and Pluto's ability to break sections of the notebook up into 'slides', it is possible to create a 'front panel' slide with user controls and data plots and a 'backend' slide that contains the code to be executed, similar in fashion to LabVIEW's front panel and block diagram Virtual Instrument structure. We chose to use Pluto because it requires minimal programming knowledge and handles





UI creation for the user, such that it is relatively accessible to scientists who may not have extensive experience programming GUIs. Thus, although we note Pluto is very limited as a GUI tool, we suggest it's simplicity and ease of use could make it useful as an open-source replacement for LabVIEW in the limited set of circumstances were only a simple GUI is needed (such is the case here), especially if continued development targets this application. A screenshot of the GUI can be found in Fig. S4.

To the best of our knowledge, this is the first example of and instrument GUI created using Pluto.

## 3 Instrument performance

In this section, we describe the preferred calibration method for the SiMPLE-PAS, which is to calibrate with $NO_2$ against a cavity ringdown spectrometer (CRDS), although we note a standard mixture of $NO_2$ could be used in labs where CRDS is not available. The accuracy of the instrument and calibration are validated by measuring the absorption cross section of ozone and

comparing measurements of AAE of laboratory-generated biomass aerosols to measurements from the previously published MultiPAS-IV instrument. Precision and detection limits are examined using Allan deviation and standard deviation analyses on blank data consisting of measurements of pure $N_2$.

### 3.1 Calibration

#### 3.1.1 Measurement of the resonant frequency

Before a measurement is made using PAS, the resonant frequency $f_r$ of the sample cell must be measured. We measure the resonant frequency prior to calibration, prior to the start of any measurement cycle, and whenever changing gas composition (e.g. nitrogen bath gas to oxygen bath gas or nitrogen bath gas to air bath gas). Typically, groups have either (1) scanned the laser frequency around the expected frequency, or (2) introduced white noise or a very fast frequency chirp into the sample cell using an integrated speaker. (e.g. Lack et al., 2006; Fischer and Smith, 2018a) In the first case, the sweep may take 5 minutes

to do a minimum number of frequencies required to determine the resonant frequency, and many minutes or hours to perform a sweep of the full frequency space of the cell. The second option is quicker, but requires additional hardware and software functionality and requires an additional port in the PAS cell through which the sound can be introduced, which may degrade the quality factor of the resonator.

For the SiMPLE-PAS, we desired a quick and repeatable method of measuring the resonant frequency that did not add to

the bill-of-materials cost of the PAS and did not require additional ports in the narrow PAS resonator. We therefore developed a novel third approach to measuring $f_r$ that is very fast like the chirp method but eliminates the need for extra hardware and extra ports in the sample cell. To measure the resonant frequency of the SiMPLE-PAS, we increase the gas flow from the nominal 300 sccm to roughly 1000 sccm. The higher flow rate leads to turbulent flow in the PAS, which creates broad-spectrum acoustic noise inside the sample cell. Sound at the resonant frequency of the cell will be amplified, leading to a peak at $f_r$ in

the FFT (similar to blowing air across a pipe or bottle). With the flow rate increased, we collect 8 waveforms and co-add the frequency spectra from the FFT of each waveform. The resulting spectrum is smoothed with a rolling average and then $f_r$ is





determined by fitting a Gaussian to the resonant peak using partial least squares regression, wherein the center of the Gaussian represents $f_r$. This approach is fast (approximately 10 seconds for a full frequency spectrum) and does not require additional hardware or ports. It can be performed manually, e.g. by adjusting a rotameter or swapping in a different orifice, or may be

automated by, e.g., using a software-controlled mass flow controller. To the best of our knowledge, this is the first time this approach has been described. A typical frequency spectrum and fit from this approach is presented in Fig. S5. The spectrum shown here was acquired in dry nitrogen at 22°C, and we measured a resonant frequency of $f_r$ = 1609 Hz. The nominal length of the SiMPLE-PAS resonator is 100 mm, and the resonator used in this work was measured as 101 mm, not including the end correction ($\Delta L$), which increases the effective length of the resonator. Although it has been typical to consider a PAS

resonator to be an open- or closed-tube resonator and apply an end correction of $\Delta L = 0.6r$, where $r$ is the radius of the cell (Miklós et al., 2001), we argue that the end of the SiMPLE-PAS resonator is more similar to a flanged opening than an open tube. We therefore apply an end correction of $\Delta L = 2 \times 0.85r$ on each end of the resonator, giving an effective length of $L_{eff} \approx 101\,\text{mm} + 2 \times (0.85 \times 2.3\,\text{mm}) \approx 105\,\text{mm}$. Assuming a speed of sound of 350 m/s for nitrogen, we calculate an expected resonant frequency of $f_r \approx 1660$ Hz. This is higher than the measured value of 1609 Hz, and would correspond to

an effective resonator length of approximately 108 mm. We have also measured the resonant frequency in $O_2$, and obtained a valued of 1514 Hz, which again is lower than the expected value of 1560 Hz. The measured value (1514 Hz) again corresponds to an effective resonator length of 108 mm. We hypothesize the difference could be due to either non-ideal resonator behavior resulting from the buffer volumes. This could potentially be minimized by increasing the diameter of the buffer volumes, which would more effectively simulate open end conditions for the central resonator, although we have not tested this. However, we

note that this error does not affect the accuracy of absorption measurements.

### 3.1.2 Calibration

Multiple methods of calibrating aerosol PAS instruments have been discussed in the literature, with the most common being calibration to mixtures of $NO_2$ or $O_3$ or calibration with an absorbing aerosol such as nigrosin or Regal Black (e.g. Lack et al., 2006; Zhang et al., 2016; Fischer and Smith, 2018b; Davies et al., 2018; Yu et al., 2019; Schnaiter et al., 2023). For gases,

these mixtures may be standard mixtures of known concentration, from which the absorption coefficient can be calculated, or may be measured directly using cavity ringdown spectroscopy (CRDS) or UV-visible spectroscopy; for aerosols, it is necessary to know the size distribution, number density, and refractive index of the aerosol being measured. Therefore, gas-phase measurements are generally simpler. To minimize error, it is desirable to measure the absorption coefficient directly at levels relevant to aerosol measurements, which CRDS is capable of; CRDS is also notably a calibration-free method, assuming the

relative ringdown times can be measured accurately. (Fischer and Smith, 2018b; Wada and Orr-Ewing, 2005) During laboratory experiments (ozone measurements, Allan/standard deviation analysis) we calibrated using a mixture of $NO_2$ in $N_2$ and measured the absorption coefficient of the mixture using CRDS. Briefly, the lab-built CRDS is composed of a 26-cm optical cavity bound on either end by a low-loss plano-concave cavity ringdown mirror ($R = 0.99993$, 25 cm R.O.C., 25 mm diameter; FiveNine Optics). A 446 nm diode laser (Coherent OBIS 445LX, 75 mW) is passed through an optical isolator (Thor Labs)

and coupled into the cavity, and an avalanche photodiode (Thor Labs APD130A) is used to detect light leaking through the





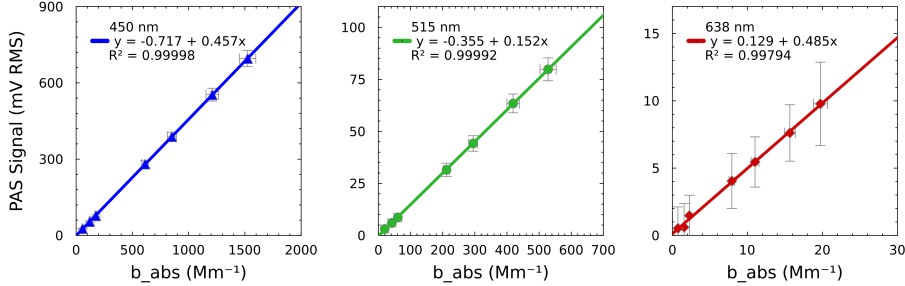

**Figure 4.** Calibration of the SiMPLE-PAS. Errors bars for $y$ represent 1 standard deviation of the PAS signal and background propagated with the uncertainty on the ratios of the $NO_2$ absorption cross sections at each wavelength ($\lambda_{PAS}$/446 nm) and error bars for $x$ are 1 standard deviation of the CRDS signal propagated with the uncertainty on flow rates and $R_L$ for 2-minute averages ($x$). The points from 0-2 $Mm^{-1}$ for the red channel highlight the 2-minute limit of detection for the SiMPLE-PAS, which is 1.0 $Mm^{-1}$ for the red channel.

back mirror. A USB oscilloscope (Pico Technology PicoScope 2000) is used to digitize the signal and sent to a custom data acquisition program written in Julia/Pluto, which collects 16 ringdown events per second, fits each, and records the average; data were acquired at 0.2 Hz to match the sampling rate of the PAS. A purge of 11.5 sccm nitrogen was maintained over each mirrors during $NO_2$ measurements using a critical orifice (Lenox Laser) to limit the flow.

During calibration, a 50-ppm mixture of $NO_2$ in $N_2$ (Airgas) was diluted with additional $N_2$ from a nitrogen generator (Vici DBS Whisper 0-120). Nitrogen flows were controlled by a mass-flow controller (Alicat Scientific) and held at 3300 sccm while $NO_2$ flows were varied using a needle valve in the range of 1-20 sccm. The two flows were combined using a stainless-steel T and allowed to mix in a 1-m length of 1/4-inch O.D. PTFE tubing. After mixing, 300 sccm of the mixture was split to the PAS while the remainder was sent to waste. The mixture passed through the PAS and was fed to the CRDS using a

half-meter length of PTFE tubing, where it was diluted by the nitrogen purge flow on the front cavity mirror, supplied by the same nitrogen generator used for dilution. At the beginning of the calibration, a higher concentration (approximately 25 ppm) of $NO_2$ was flowed through the experimental apparatus for several minutes to passivate all surfaces. Then, the $NO_2$ flow was reduced to approximately 20 sccm for the first measurement and then reduced successively until (typically) 5 concentrations were measured. Each concentration was measured for approximately 10 minutes. The ringdown time measured by the CRDS

was converted to the absorption coefficient for the mixture $b_{abs-446}$, which was used to calculate the absorption coefficient for each PAS wavelength using the ratio of the absorption cross sections at each wavelength. (Bogumil et al., 2003)

     Figure 4 shows the calibration curve for each wavelength. All wavelengths were measured simultaneously, which caused very high signal at the blue wavelength and very low signal at the red wavelength due to differences in the absorption cross section at each wavelength. This highlights the dynamic range of the PAS, which can measure from several $Mm^{-1}$ up to

over 1500 $Mm^{-1}$. Typically, the slope of a PAS calibration curve represents the cell constant of the PAS, such that the same slope should be measured for all wavelengths, assuming the signal has been normalized to incident power. Here, we choose to prepare separate curves for each wavelength instead of normalizing the power, and thereby eliminate the cost of a power meter from the PAS design. Therefore, the slope of each curve here represents the $P_0 \times C_{cell}$ (Eq. 1). We observe similar sensitivity



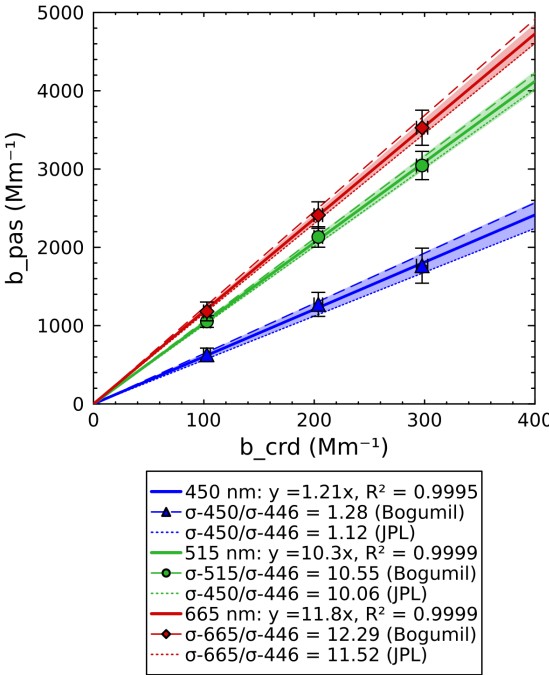

**Figure 5.** Validation with ozone showing the ozone absorption coefficient measured by PAS at each wavelength (abscissa) vs ozone absorption cross section measured at 446 nm using CRDS (ordinate). The slope of each solid line represents the ratio of $b_\lambda/b_{446}$ as measured by PAS, while dashed lines represent the expected ratio from literature values. Error bars represent the propagated uncertainties for the analysis and are discussed in the supplemental information.

for the green and red channels, but see about 3 times lower sensitivity for the green channel, which can be attributed to the

lower power output from the green laser, which operates at about 1/3 the power of the other two lasers. The points in Fig. 4 represent 2-minute averages of both the PAS and CRD measurements, with $y$-error bars representing the propagated error for each instrument, as described in the supplemental information. Although we typically measure only 5 concentrations during calibration, we show 7 here to span a larger range and highlight the ability of the PAS to measure values near the lower limit of detection (discussed further below). The data in Fig. 4 also demonstrate excellent precision, with $R^2 = 0.9999$ or greater for

the blue and green wavelengths and 0.9979 for the red. The reduced precision for the red is due to the inclusion of 3 points close to the detection limit of the SiMPLE-PAS, which we would not typically measure in a normal calibration. When these points are excluded from the calibration, we find $R^2 > 0.9999$. Since the CRD was not available for the field experiments herein (i.e. G-WISE 2), we instead calibrated against a portable corona discharge ozone generator, which itself was calibrated against a CRDS instrument after the campaign. The ozone calibration procedure is similar to that described for validation, below.





## 3.2   Accuracy determination with ozone

To verify the accuracy of the PAS, we measured absorption by ozone and compared the PAS values to those measured by the 446 nm CRDS described above and literature values. A corona-discharge ozone generator (Longevity Resources) was supplied with 99.999% $O_2$ (Airgas) at a rate of 2 standard liters per minute (SLPM) and the ozone concentration was varied with the 3 lowest output settings of the generator, corresponding to the maximum we could measure without saturating the PAS DAQ card). The 2 SLPM stream of $O_3 + O_2$ exiting the generator was split via a stainless-steel T, with 300 sccm entering the PAS via PTFE tubing and the remainder going to waste. The gas stream exiting the PAS was sent to the CRDS via a 0.5 m length of PTFE tubing; no purge flow was used in the CRDS for ozone experiments. Prior to measurements, we calibrated the SiMPLE-PAS with $NO_2$ as described above. We then re-measured the resonant frequency of the cell after purging with oxygen for approximately 5 minutes, and observed that the resonant frequency decreased by approximately 1000 Hz, consistent with theoretical expectations for pure oxygen. After measuring the resonant frequency, we flooded the cell with a high concentration of ozone for several minutes to passivate all surfaces before lowering the concentration to the third lowest setting on the generator and beginning measurements and subsequently decreasing the concentration for successive measurements. Oxygen was chosen as the diluent to eliminate error associated with photodissociation of oxygen and inefficient energy transfer to the bath gas. The use of oxygen (or air) as the bath gas has been shown to effectively eliminate this error. (Fischer and Smith, 2018b; Davies et al., 2018; Cotterell et al., 2019, 2021) We verified this by cycling each wavelength on and off and looking for changes in the other wavelengths, as described by Fischer and Smith (2018b), and saw no detectable differences.

Figure 5 shows the results of the ozone experiment. The data are plotted with 2-minute averages of the PAS-measured values on the $y$-axis and the CRDS values at 446 nm on the $x$-axis. When plotted this way, the slope of the line of best fit represents the ratio of the absorption cross sections at each PAS wavelength vs. the value for 446 nm. The solid lines in Fig. 5 represent the line of best fit (with the intercept forced through 0), while the dashed lines represent the lines expected based on literature values. For comparison, we show the values of Bogumil et al. (2003) (for 293 K) and the NASA Jet Propulsion Laboratory (JPL) recommendation (for 293-298 K) from their 2020 report on photochemical data for atmospheric studies. (Burkholder et al., 2020) Since the data of Bogumil et al. (2003) are higher resolution than the linewidth of our lasers, we take the average over the emission peak for each of the SiMPLE-PAS/CRDS lasers and report that on the plot in Fig. 5. The JPL recommendation are provided at lower resolution than our lasers, so we report the value closest to each of the SiMPLE-PAS/CRDS wavelengths. The error bars in Fig. 5 are described in the Supplemental Information and encompass the uncertainty from the calibration with $NO_2$ and indeterminate error in the ozone measurements, while the shaded regions represent the 95% confidence interval on the slope of the regression. We measure values of $1.21 \pm 0.1, 10.3 \pm 0.3,$ and $11.8 \pm 0.3$ for $b_\lambda/b_{446}$, where $\lambda = 450, 515,$ and 665 nm, respectively. We find no significant differences between the values of Burkholder et al. (2020) and the SiMPLE-PAS data at the $\alpha = 0.05$ confidence level, and further find the errors on individual points overlap with both literature sources. We therefore conclude the SiMPLE-PAS calibration is accurate with the precision of the instrument and uncertainty in literature values. We further note the implication of no (or at least minimal) loss of ozone and/or $NO_2$ to the plastic cell walls, which



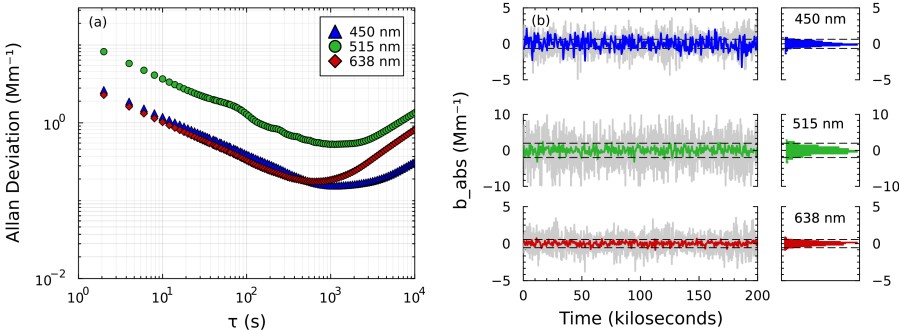

**Figure 6.** Detection limits of the SiMPLE-PAS. (a) Allan deviation results and (b) method detection limits determined from standard deviation. For (b), grey lines correspond to 2-minute averages and colored lines (and histograms) correspond to 10-minute averages. Dashed horizontal lines correspond to the $2\sigma$ limit of detection based on the 10-minute averages.

was a hypothetical concern when developing the SiMPLE-PAS. We again see high $R^2$ values, ranging from 0.9996 to 0.9999 depending on wavelength, that highlight the precision and linearity of the SiMPLE-PAS.

## 3.3 Limits of detection

To determine the limit of detection for the SiMPLE-PAS, the cell was purged with 300 sccm dry, filtered nitrogen, which was supplied by a nitrogen generator (Vici DBS Whisper 0-120) through a flow-limiting orifice. The PAS was turned on and allowed to warm up with a nitrogen purge overnight prior to data collection. After warm up, the signal for each wavelength was recorded for a period of 72 hours (3 days). The data were mean centered and divided by the sensitivity coefficient ("calibration constant") for each wavelength to convert the data to units of $\mathrm{Mm}^{-1}$. The data were then treated in two separate ways to estimate the limit of detection for the SiMPLE-PAS: (1) the absorption coefficient values were used to compute Allan deviation for the dataset and (2) the data were averaged in 2-minute and 10-minute bins and drift was removed from the data treating every fifteenth (for 2-minute data) or third (for 10-minute data) data point as a background and subtracting it from the following 14 or 2 points, respectively, and the standard deviation was computed for the entire dataset.

Allan deviation has been widely used to determine detection limit of PAS instruments. As discussed elsewhere, Allan deviation provides an estimate of the instrument's *ultimate detection limit*. (Giglio et al., 2016) Fig. 6A shows the results of our Allan deviation analysis for the SiMPLE-PAS, which look typical for Allan deviation. Allan deviation is similar to standard deviation, but isolates the effects of random noise and long-term drift, while standard deviation does not. (Langridge et al., 2008) The $x$ axis in Fig. 6A represents different averaging times, $\tau$, and the $y$ axis shows the Allan deviation for each averaging time. The left portion of each trace shows a negative slope, which represents increased precision due to cancellation of random noise as averaging time is increased. The right portion of each trace exhibits a positive slope, which represents the decrease in precision associated with long-term instrument drift. The $x$ value corresponding to the minimum deviation for each trace represents the optimum averaging time for that wavelength, with longer times representing a more stable instrument, while



the $y$ value represents the detection limit expected at that averaging time. Here, we observe good stability at all wavelengths with the blue and green wavelengths reaching a minimum deviation at averaging times of approximately 1110 and 1180 s, respectively, while the red exhibits lower stability with an optimum averaging time of approximately 680 s; it is unknown why the red wavelength is less stable. The stability of the SiMPLE-PAS is therefore very competitive when compared to other PAS instruments. For example, Fischer and Smith (2018a) show inflection points at near 1000 s for the 4 channels of their MultiPAS-IV multi-pass, 4-wavelength laser PAS; Schnaiter et al. (2023) show turn over points between 2000 and 3000 s for their single-pass, 4-wavelength laser PAS; Kapp et al. (2019) show an optimum averaging time of approximately 1000 s for their single-wavelength LED PAS; and Cui et al. (2025) show a relative short optimum averaging time of 48 s for their 3D-printed PAS. Likewise, the ultimate detection limits deduced from Allan deviation for the SiMPLE-PAS are very competitive compared to other PAS instruments. We observe detection limits of 0.15, 0.50, and 0.17 $\text{Mm}^{-1}$ at 450 nm, 515 nm, and 665 nm, respectively. Fischer and Smith (2018a) show Allan deviation based detection limits 0.05 to 0.1 $\text{Mm}^{-1}$, Schnaiter et al. (2023) report an ultimate detection limit of 0.09 $\text{Mm}^{-1}$ for their 4 wavelengths; Linke et al. (2016) report detection limits of 3.7-12 $\text{Mm}^{-1}$ for their 3-wavelength PAS; Yu et al. (2019) report detection limits in the range of 0.20-0.90 $\text{Mm}^{-1}$ for their differential, multiwavelength PAS (RGB-DPAS); Kapp et al. (2019) report a detection limit of 3 ppb NO2, or roughly 5 $\text{Mm}^{-1}$ at the central wavelength of the LED (405 nm) assuming normal temperature and pressure (NTP) (Bogumil et al., 2003); and Cui et al. (2025) report an ultimate detection limit of 0.33 ppm $CO_2$ for their multipass, 3D-printed PAS, or roughly 0.05 $\text{Mm}^{-1}$.

Allan deviation likely overestimates an instrument's limit of detection (i.e. yields a better/lower limit of detection than is practically possible). Moreover, IUPAC defines detection limits in terms of standard deviation, not Allan deviation. (Kaiser and Chalk, 2019) Many groups have begun to report detection limits based on standard deviation along with their Allan deviation results, often using the 2-$\sigma$ standard that has become common in the field of aerosol PAS. When reporting standard deviation, groups often mimic their sampling routine by recomputing a background at defined time intervals, as described above. Standard deviation therefore provides more realistic measure of an instrument's practical detection limits, which we here term the method detection limit (MDL) because it incorporates both instrumental and methodological considerations. (Nakayama et al., 2015; Fischer and Smith, 2018a; Schnaiter et al., 2023) When treating the data in this way, we observe 2-$\sigma$, 2-minute lower limits of detection of 1.2, 3.6, and 1.0 $\text{Mm}^{-1}$ and 2-$\sigma$, 10-minute detection limits of 0.63, 2.0, and 0.55 $\text{Mm}^{-1}$ for the blue, green, and red channels, respectively. This is somewhat worse than the 2-$\sigma$, 2-minute detection limits reported by Fischer and Smith (2018a) (0.61-0.75 $\text{Mm}^{-1}$, dependent on wavelength) and Schnaiter et al. (2023) (0.28 $\text{Mm}^{-1}$). Although Schnaiter et al. (2023) do not report the power of their lasers, the manufacturer used there offers the lasers used in powers that are several times up to almost 10 times greater than the power of the lasers used in this work, which likely accounts for their superior detection limits. And notably, the SiMPLE-PAS has better detection limits than other low-cost PAS instruments. Keeratirawee et al. (2022) report a 3-$\sigma$ detection limit of 100 ppbv $NO_2$ for their low-cost circuitry and a standard pas cell, which was determined by measurement of $NO_2$; this corresponds to approximately 75 $\text{Mm}^{-1}$ at their measurement wavelength of 450 nm after conversion to the 2-$\sigma$ level, assuming NTP, although no averaging time is reported. Kapp et al. (2019) report a 6-$\sigma$ detection limit of 200 ppb $NO_2$ for the LED-based PAS, also determined by measurement of $NO_2$; this




corresponds to approximately 50 $\mathrm{Mm}^{-1}$ at their measurement wavelength of 405 nm, again after conversion to the 2-$\sigma$ level
and assuming NTP, although no averaging time is reported. (Bogumil et al., 2003) Haedrich et al. (2025) report a black carbon
(BC) detection limit of 10.5 µg/m$^3$ BC for their low-cost QE-PAS at a 1-s integration time; assuming the standard BC mass-
absorption coefficient of 7.5 $\mathrm{m}^2\mathrm{g}^{-1}$ and an improvement in the detection limit that scales as $1/\sqrt{N}$, this would correspond
to approximately 7 $\mathrm{Mm}^{-1}$ for a 2-minute, 2-$\sigma$ detection limit. (Bond and Bergstrom, 2006) The SiMPLE-PAS therefore has
better detection limits than other low-cost instrumentation and is competitive with the most sensitive PAS instrument in the
literature, including multi-pass instruments. The detection limits observed here are suitable for detection of ambient aerosols
except in very clean environments and more than suitable for measurements of lab-generated aerosols.

### 3.4 Application to aerosols: Deployment at G-WISE 2

As an initial test of instrument performance and demonstration of an ability to measure aerosols, the SiMPLE-PAS was de-
ployed to the second Georgia Wildland Fire Simulation Experiment (G-WISE 2) laboratory campaign at the U.S.D.A. Forest
Service Prescribed Fire Science Laboratory (U.S.D.A. Forest Service Southern Research Station, Athens, GA, United States)
in April 2025. This experiment was a follow up to the first G-WISE campaign and the experiments and results will be de-
scribed in detail in forthcoming publications. (Saleh and O'Brien, 2025) The goal here is merely to compare the SiMPLE-PAS
performance that that of other aerosol absorption instruments at G-WISE 2, namely the MultiPAS-IV, not to draw scientific
conclusions about the samples studied or describe the scientific basis for the experiments. Briefly, fuel bed mixtures repre-
sentative of the Southeastern United States were prepared by the US Forest Service and placed in a 1-m by 1-m square in a
sealed indoor burn room. The samples, which were composed of various compositions of pine needles, broadleaf leaves, duff,
and wood, were ignited and allowed to burn to completion such that the burn room contained all the smoke from the burned
organic matter. A copper sample line was placed directly in the burn room and, for the data presented here, pumped to the
MultiPAS-IV and SiMPLE-PAS on a shared sample line without dilution. Two homemade thermodenuders were available to
remove organics from the aerosols, one set to $100°$C and one set to $300°$C. Prior to the start of sampling, the SiMPLE-PAS
was calibrated as described above and the MultiPAS-IV was calibrated as described elsewhere. (Fischer and Smith, 2018a)

Because the sample was composed of both BrC and BC, we fit the spectrum by first using the point at 665 nm to determine
the BC component and then fit the residual to determine the BrC component. The BC component is assumed to have an AAE
of 1.0, and we calculate the BC absorption coefficient ($b_{\mathrm{BC}}$) at each wavelength ($\lambda$) from 400 to 800 nm (Moosmüller et al.,
2011; Bond and Bergstrom, 2006):

$$b_{\mathrm{BC},\lambda} = b_{\mathrm{abs},665}\left(\frac{\lambda}{665}\right)^{-1.0} \tag{3}$$

We then calculate the AAE for BrC (AAE$_{\mathrm{BrC}}$), from the residual at the blue and green wavelengths:

$$\mathrm{AAE}_{\mathrm{BrC}} = \frac{\log\left(\frac{b_{\mathrm{abs},450}-b_{\mathrm{BC},450}}{b_{\mathrm{abs},515}-b_{\mathrm{BC},515}}\right)}{\log\left(\frac{515}{450}\right)} \tag{4}$$




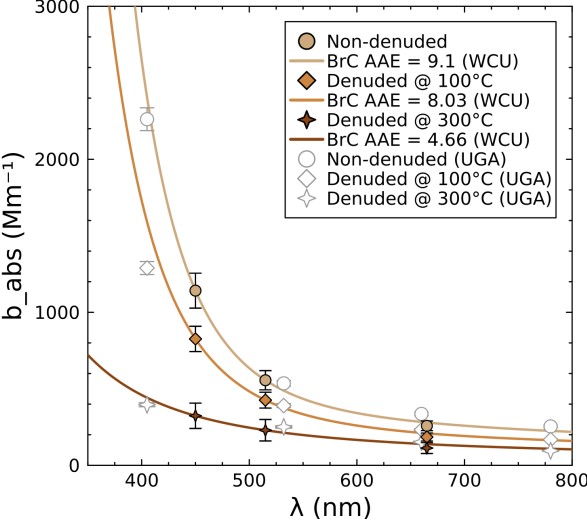

**Figure 7.** Comparison of absorption spectra for denuded and non-denuded biomass aerosols measured by two PAS instruments. Colored points represent the SiMPLE-PAS measurements while empty markers represent the MultiPAS-IV measurements. Lines are the fitted aerosol absorption spectrum based on only the SiMPLE-PAS data and do not include MultiPAS-IV data.

We use $AAE_{BrC}$ to calculate the absorption coefficient of BrC ($b_{BrC}$) at each wavelength from 400-800 nm, and calculate the final aerosol spectrum, assuming $b_{abs,\lambda} = b_{BrC,\lambda} + b_{BC,\lambda}$:

$$b_{abs,\lambda} = b_{abs,515} \left( \frac{\lambda}{515} \right)^{-AAE_{BrC}} + b_{BC,\lambda} \qquad (5)$$

Figure 7 shows aerosol absorption spectra (solid lines) for each treatment calculated from the AAE determined using the SiMPLE-PAS; MultiPAS-IV data are overlaid for comparison but are not used in the fit. It was expected that the AAE would decrease as the thermodenuder temperature increased since more volatile and semi-volatile BrC components would be removed to leave behind an aerosol that is more BC-like, and this is what was observed, although it appears that some low-volatility BrC sticks to particles even after denuding at 300°C, given that the AAE does decrease fully to 1.0. The SiMPLE-PAS was able to clearly differentiate the AAE of these treatments with the expected trend and broadly agreed with the MultiPAS-IV. However, the assumption that all absorption at 665 nm is due to BC is likely imperfect, and this, combined with extrapolation errors and calibration errors is likely to lead to discrepancies in the data. This highlights the need for broad wavelength coverage in PAS instruments whenever possible. Nevertheless, we overall conclude that the SiMPLE-PAS is able to measure the absorption coefficient of aerosols, and suggest that it is likely more accurate than other lower cost methods such as aethalometry, which is known to suffer from memory affects that cause inaccuracies in both $b_{abs}$ and the retrieved AAE values.

To further compare the SiMPLE-PAS and MultiPAS-IV measurements, we bin the data for the full experiment into 60-second averages for both instruments and use the SiMPLE-PAS AAEs to calculate the absorption at 406, 532, 662, and 782 nm (the



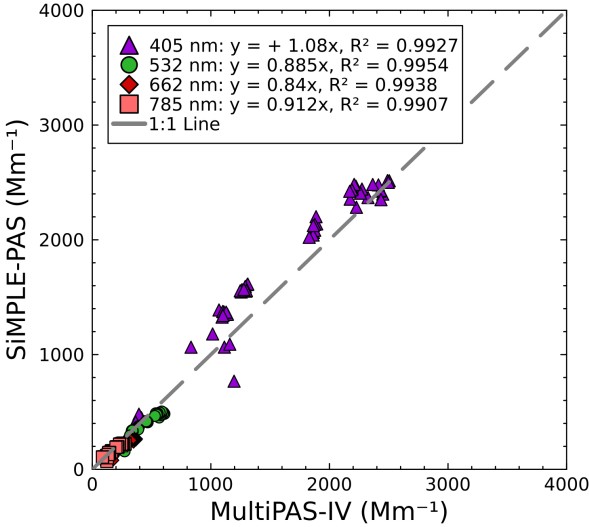

**Figure 8.** Comparison of absorption coefficients retrieved by the SiMPLE-PAS and MultiPAS-IV instruments for biomass burning aerosols.

MultiPAS-IV measurement wavelengths). In Fig. 8, we show these values plotted against the MultiPAS-IV values measured at each wavelength. We see that most points tend to cluster around the 1:1 line with $R^2$ values in the range of 0.990-0.995 (depending on wavelength, as shown by series colors in the plot), indicating that the SiMPLE-PAS agrees with the MultiPAS-IV. We observe deviations from a 1:1 relationship ranging from -15% to -9% for the IR, red, and green wavelengths and +8% for 406 nm. For the first three, we suggest this error is consistent with determinate error in the calibration with ozone, which

could have been greatly reduced by calibrating against the CRDS had it been available. For the case of 406 nm, we see a positive error. We attribute this to errors in extrapolating the SiMPLE-PAS AAE from 450 nm to 406 nm under high-AAE conditions where the wavelength dependence is very steep and therefore harder to predict via extrapolation. This analysis highlights the uncertainty involved in extrapolation and therefore need for wide spectra coverage in PAS measurements, although it has been common in the PAS literature to extrapolate from 2 or even 1 wavelength. Despite these errors, we again note that the

points overall tend to cluster around the 1:1 line and therefore believe these data demonstrate good agreement for absorption measurements of biomass aerosols.

Finally, Fig. 9 shows a comparison of the AAE values measured by the SiMPLE-PAS and MultiPAS-IV for the 60-second averages. For these data, we calculated $\text{AAE}_{\text{BrC}}$ for each instrument per Eq. 4, using the wavelength pair 450-515 nm for the SiMPLE-PAS and 406-532 nm for the MultiPAS-IV; for the gold points we calculated $b_{\text{BC}}$ using the red channel for both

instruments (Eq. (3)), while for the grey points we calculated it using the red channel for the SiMPLE-PAS and NIR channel for the MultiPAS-IV. We see from the grey points in Fig. 9 that the instruments produce different AAE values when different assumptions about the amount of BC are made. We recognize this as an inherent limitation of the SiMPLE-PAS, and any other PAS or similar instrument without NIR wavelength coverage, and suggest it is important to consider these limitations when interpreting aerosol absorption data. Notwithstanding, we see generally good agreement between AAE values measured by



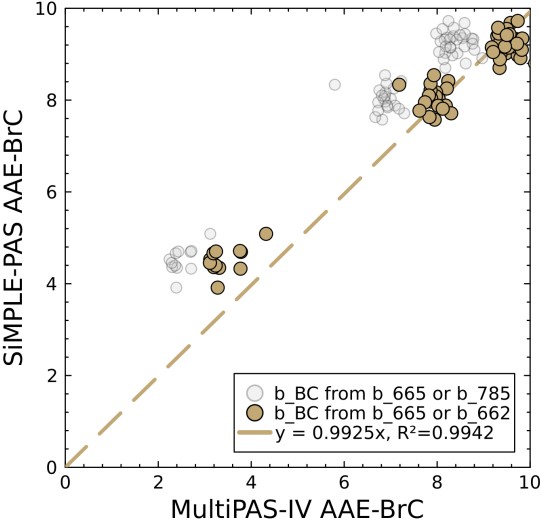

**Figure 9.** Comparison of SiMPLE-PAS $AAE_{BrC}$ values to those from the MultiPAS-IV for the wavelength pairs 450-515 nm (SiMPLE-PAS) and 405-532 nm (MultiPAS-IV). Gold points show values calculated using the red wavelength from both instruments for determination of $b_{BC}$ (Eq. (3)) while grey points use the red wavelength for the SiMPLE-PAS and NIR for the MultiPAS-IV.

the two instruments when considering the gold points (i.e. when the same assumptions about contributions from BC are used for both instruments), with the line of best fit yielding a slope of 0.99 and a correlation coefficient of 0.99. We see deviation from the 1:1 relationship in lower-AAE samples, i.e. those that were thermodenuded at 300°C. It remains unclear why the two instruments measure different AAEs for these samples, although we suggest one explanation could be an increased dominance of absorption by nitroaromatic compounds in low-AAE spectra. Since BrC with a high concentration of nitroaromatics tends to

be low-volatility and exhibit a relatively low AAE and a high absorption coefficient, this would be consistent with removal of only high-volatility, high-AAE BrC in the thermodenuder, while the low-volatility nitroaromatics are left behind. This is also consistent with the fact that the AAE does not decrease fully to 1.0 after denuding. Such compounds preferentially absorb blue light and therefore would increase absorption in the 450 nm channel of the SiMPLE-PAS while affecting the 406 nm channel of the MultiPAS-IV to a lesser extent (Singh et al., 2021). This in turn would lead the blue-green AAE values retrieved by the

SiMPLE-PAS to be higher than those retrieved by the MultiPAS-IV. We note that any instrument for *in situ* aerosol absorption measurements with a blue channel and without a violet/UV channel would be similarly affected, and that instruments with a violet/UV channel and without a blue channel may erroneously ignore nitroaromatic absorption. In fact, this 'feature' has been used to track the concentration of nitroaromatic compounds using the ratio of the 370 nm and 470 nm channels in aethalometer measurements (Barreira et al., 2024). Thus, we believe the differences at low-AAE values are due to real differences in the

sample and not measurement error, a therefore conclude the SiMPLE-PAS accurately measures $AAE_{BrC}$, especially because we see such good agreement at other values.



# 4 Summary and conclusions

The goal of this work was to put forth a design for a PAS that could be built with low-cost, readily accessible components, has detection limits suitable for measurement of both ambient aerosols and lab-generated aerosols, uses multiple wavelengths to determine AAE values, and has a total bill-of-materials cost on the order of hundreds of dollars. To that end, Fig. S6 shows a full bill of materials for the SiMPLE-PAS with costs for each component as of the time of this writing. The total cost is roughly $500, which appears to be below the cost of any PAS instrument published to date. We give suggestions to reduce the cost further, to under $200, noting that the two precision windows used in the SiMPLE-PAS account for almost half the cost of the instrument and are likely overkill for most situations. To the best of our knowledge, this is the first low-cost, multiwavelength PAS, and the first field-deployable PAS built with FDM 3D printing. While it was anticipated that some sensitivity would be given up to achieve our goals, the hope was that it would be possible to construct an instrument capable of both field deployment for ambient aerosol measurements and deployment in laboratory campaigns. And despite it's low cost, the SiMPLE-PAS has detection limits similar to those of instruments many times its cost (we refer readers to the text above for a comparison to low-cost instruments and Table S1 from Fischer and Smith (2018a) for a more complete comparison to all aerosol PAS instruments) and outperforms other low-cost instruments in terms of detection limits. In addition to presenting low-cost hardware, we have further developed an open-source GUI for the instrument, using Julia and the Julia package Pluto, which is a novel use of these tools that are typically used only for data analysis.

Although we feel we met the design goals laid out for this project, we still see room for continued improvement. Since the work described herein, we have developed a low-cost power supply system to eliminate the need for the costly bench supplies used here and make the instrument more compact and portable. Likewise, this instrument makes no effort to monitor laser power in real-time, as many instruments do in the literature. Although this seems to have little effect on accuracy due to the stable nature of the single-pass cell, it is still desirable to eliminate this potential source of error. Therefore, we plan to develop a low-cost photodiode-based power meter to be placed at the rear of the PAS cell; this device will again utilize custom electronics and a low-cost microcontroller for data acquisition. Finally, we recognize that there are inherent limitations and risks involved in using 3D printed cells for gas-phase measurements, and note that making them gas-tight requires a not insignificant amount of time investment; therefore, we plan to explore future designs that incorporate a simple and therefore low-cost machined metal resonator with 3D-printed plastic parts for mounting windows, the preamp, etc. Such a design could eliminate the bulk of the machining operations and, especially with the recent rise in low-cost online machine shops, potentially provide a more traditional machined PAS cell with little additional cost over plastic parts, especially when the cost of researcher time is factored in.

As part of this work, we compared measurements from the SiMPLE-PAS to measurements from the previously described MultiPAS-IV instrument, such that an additional outcome is to highlight the effects of wavelength choice for aerosol PAS instruments. For example, the SiMPLE-PAS has a lower wavelength limit of 450 nm, which is dictated by the low-cost laser module used in this study. In comparison, the MultiPAS-IV has a lower wavelength limit of 406 nm. This seemingly small difference has outcomes on both the measurement methods and results. The 450 nm wavelength has the advantage that it can





be calibrated directly using either $NO_2$ or $O_3$, although this may come at the expense of greater susceptibility to interference from $O_3$, more sensitivity to nitroaromatic compounds (Fig. 9) that may affect AAE measurements, and higher uncertainty on $AAE_{BrC}$ due to extrapolation to UV wavelengths (Fig. 8). Conversely, the 406 nm wavelength is much less susceptible to interference from $O_3$, but may ignore absorption from nitroaromatic aerosols and also cannot be calibrated directly with

common gas-phase calibrants and must rely on either calibration to aerosols or indirect calibration using the ratio of powers are other wavelengths (Fischer and Smith, 2018a). In a similar fashion, the MultiPAS-IV has an upper wavelength limit of 782 nm, while the SiMPLE-PAS has a limit of 665 nm, again imposed by the low-cost laser module. The effects of this are evident in measurements of AAE (Fig. 9), wherein the SiMPLE-PAS likely overestimates contributions from BC absorption to the overall spectrum. These results reflect on all aerosol PAS instruments, not just the SiMPLE-PAS, and such factors should

be considered when interpreting PAS data. Taken together, these results suggests that it would be desirable to target PAS instruments with dense wavelength coverage in the UV-visible spectrum and measurements of $b_{BC}$ far into the NIR, since there may be not-insignificant BrC absorption even near 785 nm.

In sum, we have developed a low-cost, multiwavelength photoacoustic spectrometer that accurately measures aerosol absorption and the aerosol AAE, as demonstrated with comparison measurements of biomass aerosols with the MultiPAS-IV

instrument. In this manuscript we have attempted to discuss the strengths and weaknesses of working with such a low-cost instrument, and find the primary weakness here is the limited wavelength coverage of the low-cost lasers used but that performance is very good overall. We hope that this instrument may lower the barrier to entry for those wishing to begin PAS measurements, and further realize that it may allow widespread deployment of many PAS instruments across a study area, as has been done with other low-cost sensors like the PurpleAir-II sensors. This latter point could then allow for much larger

datasets than are currently available that may be useful for, e.g., machine learning applications and understanding hyper-local aerosol characteristics.

*Code and data availability.* Data used in plots are available at https://codeberg.org/alphonse/SiMPLE-PAS. Code and CAD models are available by request from the corresponding author.

*Author contributions.* DAF conceived of and designed the SiMPLE-PAS instrument, wrote the software, contributed to instrument assembly,
data collection, and data processing, and wrote the manuscript; AMS contributed to instrument assembly and SiMPLE-PAS data collection; CAW constructed the CRD used for calibration and contributed to PCB design and 3D printing; RPP operated the SiMPLE-PAS and MultiPAS-IV at G-WISE 2 and completed data processing for MultiPAS-IV data; ADJ conducted ozone measurements with the CRD.

*Competing interests.* The authors declare no competing interests.



*Acknowledgements.* The authors express our sincere gratitude to Rawad Saleh (University of Georgia College of Engineering), Joe O'Brien
(US Forest Service), and Geoff Smith (University of Georgia Department of Chemistry) for facilitating our measurements during G-WISE
2, and further thank Geoff Smith for sharing the MultiPAS-IV data and providing feedback on the manuscript. DAF, AMS, CAW, and ADJ
received funding from Western Carolina University. RPP received funding from the National Science Foundation, Division of Atmospheric
and Geospace Sciences under grant AGS-2134617. G-WISE 2 was funded by the Department of Defense Strategic Environmental Research
and Development Program under contract RC 24-4132 and the National Science Foundation Division of Atmospheric and Geospace Sciences
under grant AGS-2144062.



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
