# Peer review of "Development of the SiMPLE-PAS: A low-cost, 3-wavelength photoacoustic spectrometer for aerosol absorption"

_Aerosol Research, 2025_

## Author Comment (AC1)

**Scott et al 2025: Development of the SiMPLE-PAS**

Authors' Response to Referees

**Contents**

**1   Responses to General Comments of Both Referees**

We thank both referees for their reviews of the manuscript. We have copied their comments here in black text and given our responses in blue. We are also submitting a revised and reformatted version of the manuscript. We will note specific changes to the manuscript using orange text under each comment; because we present a major revision and restructuring here, we will note the position of changes by section number and paragraph number within each section of the revised manuscript and have also done our best to provide approximate line numbers of the changes in the revised manuscript. We first respond to general/shared comments of both referees in one section, followed by responses to the specific comments of each referee in subsequent sections.

**1.1 Novelty and Framing**

**1.1.1 RC1 Comments**

The manuscript describes the development of a low-cost, 3-wavelength photoacoustic spectrometer (SiMPLE-PAS). The emphasis is on engineering design choices (mechanical, electrical, and software), with some laboratory validation and a limited field deployment. While the realization of the device is technically competent, I do not find significant novelty from a scientific instrumentation perspective: the underlying working principle is that of a standard photoacoustic instrument, and the use of 3D-printed parts and consumer electronics is incremental rather than conceptually new. The authors also do not clearly articulate the specific need or scientific problem that this instrument addresses beyond low cost.

That said, the work could still be of interest to Aerosol Research if framed as a reproducible, open-source, educational, or accessibility-focused contribution. To reach that point, the manuscript requires major revision, both in structure and in content.

**1.1.2 RC2 Comments**

The manuscript describes a low-cost photoacoustic instrument, capable of measuring aerosol absorption at three visible wavelengths. The authors give a highly detailed description of the design of the instrument, its calibration against gas phase absorption, and evaluation of its detection limit. A comparison test with aerosol sample against an existing photoacoustic instrument is also presented. The performance is comparable to many previously reported PAS instruments.

However, the manuscript reads a lot more like a technical tutorial rather than a research article, and the scientific novelty is quite limited. The design seems fairly conventional, outside of demonstrating that good sensitivity can be achieved with the low-cost options and technical effort has gone to designing a custom amplifier for the microphone and ensuring that the 3D-printed parts are suitable for handling gas samples and acoustic noise.

On the other hand, the topic is quite interesting and relevant, particularly because, like the authors justify in the introduction, the option for commercial PAS instruments is currently somewhat limited. Offering a detailed instructions for a low-cost starting point using readily available parts has potential for impact. So, while I would like to support the publication of the manuscript, I question if Aerosol Research is the correct publication in this case, due to limited scientific novelty of the manuscript and since *AR* offers no options for something like technical notes or tutorial type articles. I agree with comments of RC1 (`https://doi.org/10.5194/ar-2025-31-RC1`) that there is likely a way to reformat the manuscript towards a more conventional research article, but I feel like substantially shortening it could be detrimental, since main impact seems to come from the potential reproducibility of the instrument. If the authors decide to move to this direction, I suggest preserving the details at least in a supplement.

**1.1.3 Authors' Response:**

The primary concerns of both referees seem to regard the novelty of the work and therefore its suitability for *Aerosol Research* (*AR*). **We submitted this work to the "Aerosol Measurements & Instrumentation" subject area of *AR*, which specifically requests papers on "*improved* methods and instrumentation"** (emphasis ours). So while we don't specifically disagree that this work could be considered "incremental", we do argue that improvements to methods and instrumentation are often inherently incremental and very technical in nature. We further suggest that many manuscripts about PAS instrumentation published in the past ~10 years or more have been incremental to an arguably similar

amount as this manuscript. Some manuscripts have detailed addition of more laser wavelengths (e.g. Fischer and Smith (2018a)), some have detailed development of novel PAS electronics (e.g. Keeratirawee et al (2022)), and some have detailed the characterization of new cell designs (e.g. Schnaiter et al (2023)). We did all of this in our manuscript (often at a much lower cost), in addition to presenting guidelines for 3D printing PAS cells, presenting a novel system for developing DAQ/GUI software using open-source tools, and presenting a novel method for measuring the cell resonant frequency, and we additionally present an example of how to integrate all of these into a single functional system. Each of these items comes with significant technical challenges that have not been previously addressed in the literature and which we argue add value to the field and should not be dismissed.

RC1 further states "The authors also do not clearly articulate the specific need or scientific problem that this instrument addresses beyond low cost" and is concerned that the work would not of interest to *Aerosol Research* unless "framed as a reproducible, open-source, educational, or accessibility-focused contribution." We believe this is exactly what this contribution is apologize that this was seemingly unclear. In fact, this was a reason we chose to submit to $AR$ – because the article could remain freely available and be released under a permissive license, and we do frame the instrument as an accessibility-focused contribution at several places in the manuscript. In the revised manuscript, we attempt to clarify our framing by addressing the referee's concerns about the open-source nature of the instrument, as described below. We further find ourselves in agreement with RC2 on this point, who states that some of the relevance of the instrument comes simply from the dearth of commercial PAS instruments, as discussed in our original introduction.

Again, we submitted this work to the "Aerosol Measurements & Instrumentation" subject area of $AR$, which, again, specifically requests papers on "improved methods and instrumentation" We *improved* upon prior PAS instruments by offering a design with similar detection limits and features at orders of magnitude lower cost. Further, the subject area allows for the publication of "intercomparison studies", of which we present a small intercomparison here that reveals the importance of wavelength choice in PAS measurements. We believe the instrument we describe could be useful to many in the aerosol community, and chose to submit to $AR$'s Instrumentation subject area for wide exposure in the aerosol community, the open-access nature of the journal that aligns with the goals of the project, and our interpretation that it is within the bounds of the editorial description. We feel the manuscript as a whole adds value to the literature on the topic of Aerosol PAS and therefore believe it's relevant for publication in $AR$. We hope that after revision the referees and editor(s) will agree.

**Summary of revisions regarding novelty and framing:** We are adding design files and software to the version-controlled Codeberg repository linked at the end of the manuscript. We have also left in the discussions regarding the goal of this instrument to increase accessibility to PAS and to address a lack of commercial options.

**1.2   Organization and Length**

**1.2.1   RC1 Comments**

At ~12,000 words (not accounting figures and tables), the manuscript is 4,000–6,000 words too long. Sections should be significantly shortened or moved to the supplement. Moreover, the current organization is confusing: for example, calibration methods are embedded in the Results section rather than Methods. The paper should follow standard structure before detailed discussion is considered.

**1.2.2 RC2 Comments**

I agree with comments of RC1 that there is likely a way to reformat the manuscript towards a more conventional research article, but I feel like substantially shortening it could be detrimental, since main impact seems to come from the potential reproducibility of the instrument. If the authors decide to move to this direction, I suggest preserving the details at least in a supplement.

**1.2.3 Authors' Response**

Both reviewers commented on confusing organization of the paper; we thank the reviewers for their input here. Again, we submitted this manuscript to the Aerosol Measurements & Instrumentation subject area; we apologize if we missed specific guidelines for section names and formats in $AR$'s polices, but we did not see any and therefore chose to format the paper in a way that is not uncommon for papers describing instrumentation. However, it seems both reviewers strongly dislike this format. Therefore, we revised the the manuscript to follow a more traditional Intro-Methods-Results-Conclusions format and hope this will help address the concerns of the reviewers.

Regarding length, RC1 has suggested that the paper is '4,000–6,000' words too long. We are unable to find any specific guidelines for word count in $AR$'s editorial documents, so we are unsure where this number is coming from and ask for the editor(s) and/or referee to please point us to these guidelines if we have overlooked them. Moreover, at just over 27 pages (as prepared with the LaTeX template), we believe the manuscript is comparable to other preprints submitted to $AR$ in the past several months, which range in length from approximately 15 to over 50 pages, with most falling within the range of 20-30 pages. Further, we tend to agree with RC2 that extensive shortening of the manuscript would be detrimental and moreover believe it would be hard to address the other concerns raised by RC1 in a manuscript that is half the length. We therefore plan to leave the length roughly as-is in the revised manuscript, though we are amenable to specific suggestions about things that should be cut or moved to the supplemental.

**Summary of revisions regarding organization and length:** We have completed a major reorganization of the manuscript to better fit the format of a traditional scientific manuscript rather than that of an instrumentation paper but have left the length roughly as-is. Although these changes largely consisted of rearranging text and renaming sections, we have, where prudent, added or removed short passages to help with flow in the reformatted manuscript without affecting the content.

**2 Responses to RC1**

**2.1 Reproducible and open-source availability**

**RC1:** If the intention is to provide a community-sensor-type instrument, all essential resources (software, CAD files, PCB designs) must be openly available in a long-term, independent repository. Available by request is not sufficient. An assembly guide with photographs would further enhance reproducible and impact.

**Response:**

We appreciate the referee's commitment to open source here. We agree that our intent is to create an accessible design and therefore are updating the Codeberg repository linked at the end of the original manuscript with design files and software. We likewise agree an assembly guide with photographs could be useful, although we made a concerted effort to provide detailed assembly instructions in the original manuscript and also provided photos of the assembly process in the supplement to the original manuscript.

We therefore feel there is enough information in the manuscript for it to stand on its own and think an assembly guide is outside the scope of this manuscript. Further, we intend for such an assembly guide to be a living document that is continually updated (with community input) as the design and methods evolve and is thus outside the scope of what the referees are able to review as part of this manuscript. So, while we agree with RC1 here and hope to develop such a guide in the future, we feel it is a separate entity from this manuscript. Any guide created in the future would be available via our Codeberg repository that is already linked in the paper.

**Relevant revision:** We have changed the 'Code and data availability' section from:

Data used in plots are available at `https://codeberg.org/alphonse/SiMPLE-PAS`. Code and CAD models are available by request from the corresponding author.

to:

Data used in plots, code, and CAD models are available at `https://codeberg.org/alphonse/SiMPLE-PAS`.

**2.2   Calibration and Evaluation**

**RC1:** The methods used to calibrate and evaluate the device are not sufficiently thorough or clearly explained. In particular, the lack of a conventional field evaluation with side-by-side reference instruments is unfortunate, as this is typically the best way to obtain a general understanding of the instrument response characteristics (e.g. susceptibility to varying relative humidity and temperature, long-term drift, influence of aerosol composition etc). I encourage the authors to seriously consider whether such an evaluation could be arranged. The comparison of Ångström exponents with denuded and non-denuded samples is not without interest, but the measurement arrangement introduces multiple sources of uncertainty that make explicit conclusions about the device performance difficult to draw.

**Response:**

The referee says there is no 'conventional field evaluation'. It seems their primary concerns are related to (1) changes in environmental conditions that may lead to long-term drift and (2) how the instrument responds to changes in aerosol composition. We thank the referee for their concern here, but find it unclear how the deployment described in the manuscript (G-WISE 2) is insufficient.

- To the first point, the conditions at G-WISE are very similar to many field campaigns (many instruments co-located in a small warehouse area with no climate control and open to the elements on one side). The instrument therefore experienced significant changes in humidity, temperature, sun exposure, etc., and was subject to significant external noise from other equipment and jostling from other scientists in the small facility. As noted in the manuscript, a description of the facility and campaign as a whole will be published in forthcoming manuscripts and was previously published for the very similar G-WISE 1 campaign cited in the manuscript, so we have chosen not to go into further detail here.

- To the second point, we note that we have much more control over the sample than one would have in a 'conventional' field campaign (e.g. one measuring only ambient aerosols) and therefore know more about what we should expect in the sample. We thus disagree with RC1 here and find that the campaign described herein explicitly provides an ability for us to control the sample that provides a *better* validation of the instrument than we would have gotten in a field campaign in which we have little to no *a-priori* knowledge of the sample.

The referee is also concerned that we did not do an 'evaluation side-by-side with reference instruments'. Again, we appreciate the referee's intent but are unclear what is meant here. To the best of our knowledge,

there is no generally accepted reference instrument for measuring aerosol absorption. To the contrary, many of the commercial instruments available for this purpose (e.g. aethalometers) are well-known to be prone to interference from other factors and frequently show poor agreement in both absorption coefficient and AAE when compared to PAS instruments, as discussed briefly in the original manuscript (e.g. Weingartner et al 2003, Fischer and Smith (2018a), Schnaiter et al (2023), Yus-Díez (2023), Yus-Díez et al (2025), and more). We therefore chose to compare to a previously validated/published PAS instrument as a direct comparison and observed relatively good agreement. It is also unclear to us how the measurement arrangement at G-WISE 2 introduces unusual uncertainty – we send the same samples of varied compositions to two instruments on the same sample line without dilution and directly compare the results. The two instruments measure the same sample using the same widely proven technique and should therefore produce the same results.

The referee suggests we attempt to arrange a field campaign for evaluation of the instrument in a field environment. This was our purpose in deploying the instrument at G-WISE 2, which was chosen in part for the reasons described directly above. We hope that the clarification provided there is enough to address the referee's concerns. Beyond that, it is far outside our financial abilities and time constraints to arrange a long-term field study with co-located instrumentation within the reasonably near future. We believe the G-WISE data serves this purpose and wish that additional field campaigns will be the topic of subsequent manuscripts related the the SiMPLE-PAS. Again, we hope the additional clarification provided herein is enough to address the referee's concerns.

**Relevant Revision:**

- Section 3.5, paragraph 1 (line 528): Add references supporting poor agreement between PAS/PTI and aethalometer measurements to existing sentence at end of paragraph. Some references describe the theoretical basis for the issues while others provide examples of poor agreement.

**RC1:** It is unclear why the calibration is conducted first with $NO_2$ and then again with $O_3$ using the corona discharge and prior CRDS calibration.

**Response:**

We understand that the two methods of calibration are less than ideal and potentially confusing. As we described in the manuscript, the preferred method of calibration is to use nitrogen dioxide, so this is what was done for laboratory evaluations and what we describe in the methods section. Calibration to ozone was done as a single-case in the field due to equipment and time limitations of the authors during G-WISE 2. We have moved the relevant text describing this to the G-WISE 2 section of the paper in an attempt to clarify this, although we appreciate specific suggestions of how to clarify further if needed.

**Relevant Revisions:**

- Move "Since the CRD was not available for the field experiments herein (i.e. G-WISE 2), we instead calibrated against a portable corona discharge ozone generator, which itself was calibrated against a CRDS instrument after the campaign. The ozone calibration procedure is similar to that described for validation, below." from section 2.3 to section 2.6 (paragraph 1) to highlight that ozone calibration was used only for GWISE-2.

- Section 2.6, paragraph 1 (line 385): Change "not available for the field experiments herein (i.e. G-WISE 2)," to "not available during G-WISE 2" in the same passage.

**RC1:** The manuscript states that $O_3$ is used for accuracy determination, but no quantitative metric of accuracy is provided...Why should the PAS and CRDS produce a 10x difference?

**Response:**

We thank the referee for their concern here, but to the contrary, we explicitly discuss both of these points at length in the manuscript and are not sure where the miscommunication lies. To the first question, the referee is correct that the two instruments should not be 10x apart, and we do not present this number in the manuscript! Actually, if accurate, the instruments *must* be a factor of ~12x different (at 517 nm) because each instrument measures the absorption coefficient ($b_{abs,\lambda} = N\sigma_\lambda$) at its operating wavelength, $\lambda$. Because the absorption cross section ($\sigma$) varies with wavelength, the two instruments should *not* produce the same value. Rather, the slope of the line on the plot should be equal to the *ratio of the cross sections at the operating wavelengths* for each instrument/channel. While we recognize that there are a few steps involved in interpreting Fig. 5 beyond simply looking at a 1:1 line, we chose to plot the data in the way we did because it allows us to show the raw values measured by each instrument on the x- and y- axes directly rather than needing to choose and rely on an absorption cross section from the literature. We have attempted to clarify this in the revised manuscript.

To the second point, we explicitly state in the manuscript that we are unable to differentiate our measurements from literature values within the random errors on the measurements and therefore consider the measurements accurate within random error. Alternatively, we also and discuss that there is no significant difference between our values and literature at the 95% confidence level. We do not provide a specific number beyond this because that number lies within the indeterminate error of the instrument and is therefore not readily quantifiable. In short, we are limited by indeterminate (random) error, not determinate (accuracy) error, which we described in the original manuscript and supplement.

We feel these items were explained in the original manuscript and have made only minimal changes, but are again appreciative of specific suggestions as to where the confusion results.

**Relevant Revision:**

- Section 3.3, paragraph 1 (line 444): Add "(i.e. $\sigma_\lambda/\sigma_{446}$, where $\sigma = b_{abs}/N$ if $N$ is the number density of ozone)

- Eliminate the term "accuracy" from the section headings related to ozone to avoid confusion.

**RC1:** Why is the slope of the linear fit forced through zero? Reporting $R^2 = 0.9999$ for a fit with three data points spanning 0–5000 Mm$^{-1}$ is potentially misleading. Values should be reported with realistic precision (not five significant decimals).

**Response:**

We thank the referee for their attention to detail here. We agree that we should not force through 0 in this case and have therefore re-done this analysis/figure with a floating intercept and will incorporate this into the revised manuscript. We note that this had effectively no consequence on the conclusions drawn from these data, although it did change the uncertainties on the data. We have therefore removed ribbons (which represented the 95% CI of the plot) from the plot for clarity.

The referee is also concerned that we report $R^2$ values to 5 significant digits. While we agree that this is a potentially unrealistic number of significant digits, we chose to report to 5 digits to avoid reporting a value of 1.000. Clearly, there must be some imprecision in the data and a value of 1.000 is cannot be accurate; we therefore choose to report excess significant digits here rather than stating we have perfect agreement with the fit. Where appropriate, we have reduced the number of digits on $R^2$ in the revised manuscript, but have continued to leave digits where rounding would lead to an implication of perfect precision.

We show only 3 points in this figure due to limitations on the available ranges of our ozone generator and the upper limit on the SiMPLE-PAS. As a counterpoint to the referee's suggestion that the data are misleading, we suggest that similar linearity is seen in the $NO_2$ calibration, which includes more points.

**Relevant revision:** We have updated Fig. 5 to show fits with a floating intercept. We have also reported $R^2$ to fewer digits in the figure. Finally, we have removed ribbons to avoid having the plot be too busy. Correspondingly, we have adjusted the following text:

- Supplemental Section 4, paragraph 2: Remove "We also calculate the 95% confidence interval (CI) for the slop of the regression of the PAS data vs. CRDS data as Student's $t$ value times the standard error of the predicted slope. Because the CIs of the slopes overlap with literature values, and the uncertainties on each point overlap with literature sources,"

- Section 3.3: delete ", while the shaded regions represent the 95% confidence interval on the slope of the regression."

- Section 3.3 (line 458): Change "$R^2$ values, ranging from 0.9996 to 0.9999 depending on wavelength" to "$R^2$ values, ranging from 0.997 to 0.9999 depending on wavelength"

- Section 3.3: Change "We measure values of $1.21 \pm 0.1, 10.3 \pm 0.3,$ and $11.8 \pm 0.3$ for $b_\lambda/b_{446}$, where $\lambda$ = 450, 515, and 665 nm, respectively." to "We measure values of $1.2 \pm 0.8, 10.2 \pm 3.6,$ and $12.0 \pm 1.2$ , where $\lambda$ = 450, 515, and 665 nm, respectively, and errors represent the 95\

- Section 3.3: Remove " (with the intercept forced through 0)".

- Fig. 4: Reduce significant digits in $R^2$ values for consistency with other changes.

**RC1:** The current presentation largely omits the 0–10 $\text{Mm}^{-1}$ range, which is highly relevant for outdoor measurements. Figures 4 and 5 would be more informative if e.g. replotted with log-scaled axes to highlight this range

**Response:**

We agree that this is the most important range for ambient measurements, which is why we chose to show our calibration plots – which show this range – in the manuscript instead of the supplemental and specifically collected calibration down to the several $\text{Mm}^{-1}$ range for the red channel. Unfortunately, we are limited in our ability to reach lower values for the blue and green channels by a combination of our experimental apparatus/materials and the dynamic range of the instrument. These points were discussed in the original manuscript and we believe both the calibration data and limit of detection data already presented in the manuscript clearly demonstrate the instrument's low-$\text{Mm}^{-1}$ detection limits.

The referee further suggests plotting the data in the calibration/validation plots on a log scale to better show low values. We thank the referee for pointing this out and do not disagree that the low values on the plot are hard to read for the blue and green wavelengths. However, the plot is showing the relationship between PAS signal and absorption coefficient, which is expected to be a linear relationship, per eq. 1 in the manuscript. We believe plotting the data on log scales would obfuscate this relationship and lack physical basis. We further considered adding an inset to the plots showing the values at lower absorption, but found the small size of the inset did not allow significantly greater detail to be gleaned from the plot while at the same time added complexity to an already busy plot. We have therefore instead added detail plots to the revised supplemental information that show the lower values in additional detail for Fig. 4.. We choose to add this to the supplemental instead of the main manuscript for the reasons described above, because we feel much of the relevant information can be gleaned from the equations presented on the chart, the charts that are unclear to not cover the 0–10 $\text{Mm}^{-1}$ of concern, because we find the 0–10 $\text{Mm}^{-1}$ range to already be clearly visible in the plot for the red channel, and to avoid adding length to an already long manuscript. We left Fig. 5 unchanged for the same reason that it should be a linear, not logarithmic, relationship and because there is no additional detail in the low end of the scale to reveal.

**Relevant revisions:**

- Add Fig. S8, which is a modified version of Fig. 4 showing a ~10X zoom of each plot side-by-side with the original.

- Fig. 4 caption: Add "Zoomed-in version of the plots showing low absorbance values are provided in the Fig. S8."

**3 Responses to RC2**

**RC2:** Comparing the sensitivity, for example to the MultiPAS-IV, the resonator Q and laser power seem to be on similar levels, but MultiPAS-IV uses multipass configuration, which gives an power enhancement of 30x or more, according to Fischer and Smith (2018a). Are you able to comment on what factors are contributing to the fact that the SiMPLE-PAS reaches close to same level of a detection limit with a single pass instrument? Is this mostly due to a more sensitive microphone and amplifier, or are there other contributing factors that might explain this difference?

**Response:**

We appreciate this question and apologize for our lack of explanation here. In fact, we had a full section and figure detailing this in our draft, but removed these to prior to submission to reduce length and because we believe there is enough information to cover to justify a separate manuscript. The difference here is related to the cell diameter: The MultiPAS-IV uses a relatively large diameter resonator to accommodate the multipass laser beam while the SiMPLE-PAS uses a much smaller diameter cell only slightly larger than the laser beam diameter. Per equation 2 in the manuscript, the cell constant is inversely proportional to the cross-sectional area of the cell, and it happens that this effect roughly cancels the enhancement from a multipass cell used by the MultiPAS-IV. This was also recently demonstrated, although not explicitly discussed, by Schnaiter et al (2023) and is something we hope to more fully describe in a subsequent manuscript. Notwithstanding, we added a short passage to the revised manuscript explaining this.

**Relevant revision:**

- Section 3.4, paragraph 1 (line 480)- Add "Notably, their instrument uses a multi-pass cell to increase sensitivity and thereby achieve lower detection limits, although here we show similar numbers for the single-pass SiMPLE-PAS, which we attribute to the smaller cell diameter of the SiMPLE-PAS (per Eq. 1). This is in agreement with the work of Schnaiter et al (2023),"

  **RC2:** The authors state that SiMPLE-PAS is the lowest cost PAS instrument to date, which, while likely to be true, is difficult to justify properly, as the costs are rarely addressed in scientific publications, and can be out of date relatively quickly, and comparisons of only material costs to commercial instruments is typically unfair.

**Response:**

We agree with the referee's comment about price comparisons and have therefore tried to tone down our language throughout the revised manuscript where cost of the PAS is discussed while making an effort to not lose the focus on the low-cost nature of the instrument.

**Relevant revisions:**

- Abstract (line 13): Change "the lowest-cost PAS to date" to "likely the lowest-cost PAS to date".

- Section 4, paragraph 1 (line 580): Change "the SiMPLE-PAS has detection limits similar to those of instruments many times its cost" to "the SiMPLE-PAS has detection limits approaching those of the most sensitive instruments in the aerosol literature"

**RC2:** One concern I would have when using a low-cost laser module without temperature stabilization and having left out the laser power monitoring, is the long-term drifting. That is, are they large enough to significantly affect the calibration over time. For example, fig. 6 shows that the drifts start to overcome the noise after around 30 min mark, but it is unclear how large the drifts are over the whole 3-day measurement, and if the relative change in the signal level can be considered negligible after the background subtraction.

Again, we appreciate this thoughtful question. The referee is correct that drift starts to overcome noise around the 30 min mark in Fig. 6 and in noting that we assume a constant calibration constant for these data without verifying that laser power does not drift. The main concern of the reviewer here seems to be the unmonitored laser powers, and whether they may change over time and therefore cause the calibration constant to change (i.e. $P_0$ in eq. 1 changes over time and therefore causes a change in $S$ that is not due to the analyte). More specifically, we interpret the referee's statement to mean they are concerned with our *measured* calibration constant changing over time, which in our case is $P_0 \times C_{\text{cell}}$. In response to the referee's concern, we note that Fig. 6 (a) inherently includes any drift in the laser power and therefore provides an upper bound on the detection limits that would be observed with power monitoring and subsequent data correction for a particular averaging time.

In Fig. 6 (b), we have removed drift using background subtraction. This approach is typical for recent PAS characterizations, and, although not often discussed, it typically omits any consideration of temperature-dependent drift (e.g. alignment, resonant frequency shifts, etc.), as we have done here (e.g. Nakayama et al (2015), Fischer and Smith (2018a), Schnaiter et al (2023)). The difference in our study is that we do not monitor laser power, and therefore have an additional uncontrolled factor over other studies that could lead to increased error, since laser power is resultingly rolled into our calibration. For transparency in this regard, we have prepared an alternative version of Fig 6 (b) showing the uncorrected signal over the full experiment and added it to the supplemental information. It can be seen that on any given day drift is on the order of several $\text{Mm}^{-1}$ and $< \pm 5 \text{ Mm}^{-1}$ for the blue and red channels and $< \pm 10 \text{ Mm}^{-1}$ for the green channel over the full experiment. Moreover, we see that for the green and red channels drifts correspond to a 16/8 hour (day/night) cycle that correlates to the timing of building HVAC schedules. While we cannot definitively say what the source of this drift is, we have observed similar cycles even while using temperature-stabilized lasers and attribute most of the drift to changes in alignment, thermal expansion / contraction of (especially plastic) parts, and uncorrected drifts in the resonant frequency – problems that other instruments likewise experience. It is difficult to provide a direct comparison to other PAS instruments here because the uncorrected data are not typically included in manuscripts and we ultimately think that robust methods and proper usage minimize the effects of laser power drifts and again emphasize that the drift shown in the new figure encompasses all sources of drift, not just laser power.

**Relevant revision:**

- Section 3.4, paragraph 2 (line 512): Add "But one notable difference between the SiMPLE-PAS and other aerosol PAS instruments is the lack of a correction for drifts in laser power (i.e. we do not monitor $P_0$ in Eq. 1). Thus, the detection limits presented here include drifts in the laser power, and incorporating a power monitor in the future would likely lead to lower detection limits. Nevertheless,"

- Supplemental Information: Add Fig. S7 showing uncorrected data for Fig. 6 (b).

- Fig. 6 caption: Add "Uncorrected data from (b) are shown in Fig. S7."

**RC2:** Fig. 4 caption: considering mentioning that the points in the figure are 2 min averages. This would clarify the point made in the last sentence of the caption.

**Response:**

We thank the reviewer for their suggestion regarding the figure 4 caption and have made this change in the revised manuscript. For clarity and consistency, we also updated this in the text and in section related to ozone, including Fig. 5.

**Relevant revisions:**

- Section 2.3, paragraph 3 (line 343): Add "All measurements are reported as 2-minute averages.'

- Section 2.4 (line 360): Add "Again, all measurements are reported as 2-minute averages."

- Fig. 4 Caption: Add "Points represent 2-minute average measurements of $NO_2$".

- Section 3.2, paragraph 1 (line 423): Add ", with points representing 2-minute averages of measurements for a single concentration of $NO_2$".

- Fig. 5 Caption: Add "Points represent 2-minute average measurements of a single ozone concentration."

**RC2:** Line 389: I assume this should say decreased by 100 Hz?

**Response:**

We thank the reviewer for pointing out our typo regarding the cell resonant frequency and have made the change suggested in our revision.

**Relevant revision:**

- Section 2.4 (line 353): Change "1000 Hz" to "100 Hz".

**RC2:** Fig. 5: The slope of the blue line in the figure does not seem to be 1.21. At $b_{CRD} = 100$ Mm$^{-1}$, it looks to be something around 600 Mm$^{-1}$. For clarity, I would also consider plotting, instead, the measured absorption as a function of the absorption calculated from the CRDS and the wavelength dependence, like was done in figure 4.

**Response:**

The referee is correct that the slope of the blue line in Fig. 5 is *not* 1.21, and we thank them for noticing and questioning this. The slope is $1.21 \times 5$, since the y-values were multiplied by 5 for the blue channel to more clearly show them on the same scale as the other wavelengths. The slope is therefore 6.05, which agrees well with the referee's observations. We apologize for omitting this point of clarification from the manuscript, and again thank the referee for noticing. We have attempted to clarify this in the legend, caption, and main text.

We also thank the referee for their comments about the axes in Fig. 5. We agree that the axes in Fig. 5 may be somewhat convoluted at initial glance, and considered plotting similarly to Fig. 4 as the reviewer suggests. In Fig 4, we are mapping out the relationship between the PAS signal and the absorption coefficient, so we must plot absorption coefficient in Mm$^{-1}$ on the x-axis. But in Fig. 5, we are comparing two instruments and so converting the absorption coefficient introduces additional uncertainty in the x-axis that is not absolutely necessary. As described above in response to RC1, we chose the axes shown because they allow us to plot the direct measurements from both instruments without the need to assume

an absorption cross section. This eliminates a source of uncertainty in the x-axis. For these reasons, we prefer to leave the axes as-is in the revised manuscript, but have attempted to add clarifying detail about the plot in the discussion within the main text.

**Relevant revisions:**

- Fig. 5 legend: Add "5 ×" to the blue label.

- Fig. 5 caption: Add "PAS values for the blue channel have been multiplied by 5 for clarity on the plot."

- Section 3.2: Add " (note that data from the blue PAS channel have been multiplied by 5 for clarity on the plot)" to the second sentence.

- Section 3.3, paragraph 1 (line 444): Add "(i.e. $\sigma_\lambda/\sigma_{446}$, where $\sigma = b_{abs}/N$ if $N$ is the number density of ozone)

**RC2:** Fig. 7: what is the fit here? Or is it the function calculated using eq. (3)-(5)?

**Response:**

The referee is correct that the function is calculated using eq. 3-5, and we thank them for pointing out the source of confusion. We will attempt to clarify our language in the revised manuscript.

**Relevant revisions:**

- Fig. 7 caption: Change "Lines are the fitted aerosol absorption spectrum based on only the SiMPLE-PAS data and do not include MultiPAS-IV data." to "Lines are the calculated aerosol absorption spectrum (Eqs. 3-5) based on only the SiMPLE-PAS data; MultiPAS-IV data were not used to determine the lines."

- Section 2.6, paragraph 2 (line 390): Change "Because the sample was composed of both BrC and BC, we fit the spectrum by first using the point at 665 nm to determine the BC component and then fit the residual to determine the BrC component." to "Because the sample was composed of both BrC and BC, we determine the overall spectrum by first using the point at 665 nm to determine the BC component and then use a line matching the residual to determine the BrC component. "

- Section 3.5, paragraph 1 (line 519): Change "not used in the fit" to "not used in computation of the overall spectra"

**RC2:** line 506: Should this say given that AAE does not decrease fully to 1.0.? This AAE treatment is currently a little unclear to me. It seems that the manuscript only reports $AAE_{BrC}$, but nowhere the effective AAE of the total sample. So, in the limiting case that there was only BC left after denuding, would $AAE_{BrC}$ not become undefined (i.e. the residuals in eq. 4 would be zero)? In my opinion the current treatment leaves unclear, how far from BC the sample actually is after denuding.

**Response:**

We again thank the referee for pointing out our typo on line 506 and will fix this in the revised manuscript.

Regarding the AAE treatment, we agree with the referee's comments that we report only $AAE_{BrC}$ and that we cannot conclusively say how much BC is left after denuding. As implied in the manuscript and as the referee suggests, we would expect residuals of 0 $Mm^{-1}$ in the case that only BC were present. We use the existence of any residual as evidence for some form of BrC, as discussed in the manuscript. However,

our goal here is not to thoroughly discuss the properties of the aerosol or the effects of thermodenuding but rather to provide an intercomparison between the two PAS instruments and highlight how wavelength selection affects the measurements. A full discussion of aerosol properties with respect to thermodenuding and with comparison to other instrumentation is forthcoming in future publications and we believe outside the scope of the current manuscript.

**Relevant revisions:**

- Section 3.5, paragraph 1 (line 523): add "not" in the position specified by the referee.

**RC2:** Related to the AAE treatment, I think there may be an error in eq. 5: are you not double counting $b_{\mathrm{BC},515}$ here? For example, if you were to calculate $b_{abs,515}$ using this, you would end up with $b_{abs,515} + b_{\mathrm{BC},515}$.

**Response:**

We thank the referee for their attention to detail in pointing out our typo in eq. 5. Their interpretation is correct based on how the equation is written, although that does not match how we processed the data. It should read '$b_{\mathrm{BrC},515}$' (not '$b_{abs,515}$') to correctly reflect what we did. We have updated this in the revised manuscript.

**Relevant revisions:**

- Section 2.6, paragraph 2 (lines 395-400): Change

$$b_{\mathrm{abs},\lambda} = b_{\mathrm{abs},515} \left( \frac{\lambda}{515} \right)^{-\mathrm{AAE_{BrC}}} + b_{\mathrm{BC},\lambda}$$

  to

$$b_{\mathrm{abs},\lambda} = b_{\mathrm{BrC},515} \left( \frac{\lambda}{515} \right)^{-\mathrm{AAE_{BrC}}} + b_{\mathrm{BC},\lambda}$$